# Spin occupancy regulation of the Pt *d*-orbital for a robust low-Pt catalyst towards oxygen reduction

Dongping Xue[1], Yifang Yuan[2], Yue Yu[1], Siran Xu[1], Yifan Wei[1], Jiaqi Zhang[1], Haizhong Guo [2], Minhua Shao [3] & Jia-Nan Zhang [1] ✉

Disentangling the limitations of O-O bond activation and OH* site-blocking effects on Pt sites is key to improving the intrinsic activity and stability of low-Pt catalysts for the oxygen reduction reaction (ORR). Herein, we integrate of PtFe alloy nanocrystals on a single-atom Fe-N-C substrate (PtFe@Fe$_{SAs}$-N-C) and further construct a ferromagnetic platform to investigate the regulation behavior of the spin occupancy state of the Pt *d*-orbital in the ORR. PtFe@Fe$_{SAs}$-N-C delivers a mass activity of 0.75 A mg$_{Pt}^{-1}$ at 0.9 V and a peak power density of 1240 mW cm$^{-2}$ in the fuel-cell, outperforming the commercial Pt/C catalyst, and a mass activity retention of 97%, with no noticeable current drop at 0.6 V for more than 220 h, is attained. *Operando* spectro-electrochemistry decodes the orbital interaction mechanism between the active center and reaction intermediates. The Pt $dz^2$ orbital occupation state is regulated to $t_{2g}^6 e_g^3$ by spin-charge injection, suppressing the OH* site-blocking effect and effectively inhibiting H$_2$O$_2$ production. This work provides valuable insights into designing high-performance and low-Pt catalysts via spintronics-level engineering.

The future of a sustainable energy supply needs innovative breakthroughs in the design of inexpensive and durable catalysts for efficient proton exchange membrane fuel cells (PEMFCs). Currently, platinum (Pt)-based catalysts are considered the most promising commercial catalysts for the cathodic oxygen reduction reaction (ORR) of PEMFCs, however, their high cost and limited reserves impede their large-scale commercialization[1–4]. Therefore, reducing the amount of Pt used in the catalyst without compromising the activity and durability is an urgent issue. Ordered PtM alloy nanostructures (where M is Ni[5], Co[6], or Fe[7], among other metals[8]) have been considered successful catalysts for decreasing the Pt load while increasing the catalytic activity of the ORR in acidic media. Current research shows that the high performance of the PtM alloy nanostructures is regulated by the *d*-band center theory[4], strain effect[9], size effect[10], and surface effect[11]. Despite the great progress that has been made in the

development of advanced Pt-based catalysts to improve Pt utilization and mass activity towards ORR, high activity, and long life are still challenging issues for PEMFCs because some kinetic dilemmas cannot be circumvented[7,12–14].

To achieve highly durable and active low-Pt-based electrocatalysts, several critical issues need to be addressed. (1) The Pt active sites still exhibit a tendency to strongly adsorb oxygen-containing intermediates and need higher overpotentials. (2) Their stability in acidic media is also unsatisfactory because of the H$_2$O$_2$ through the associative pathway; H$_2$O$_2$ will attack the active sites and carbon support. To address these issues, a dissociation pathway is desired to reduce the O-O bond activation energy barrier, bypassing the direct production of OOH*, and thereby avoiding the side reactions by introducing new variables (second adsorption site[15,16]). In this regard, developing structurally novel Pt sites to regulate the adsorption state

[1]School of Materials Science and Engineering, Zhengzhou University, Zhengzhou 450001, China. [2]Key Laboratory of Materials Physics, Ministry of Education, School of Physics and Microelectronics, Zhengzhou University, Zhengzhou 450052, China. [3]Department of Chemical and Biological Engineering, The Hong Kong University of Science and Technology, Kowloon, Hong Kong. ✉e-mail: zjn@zzu.edu.cn

of $O_2$ and oxygen-relevant intermediates (OOH*, O*, and OH*) is a promising way to promote ORR kinetics in practical applications. In principle, an excellent ORR electrocatalyst should have an optimal $d$-orbital electronic structure of the metal sites to achieve an ideal adsorption Gibbs free energy to facilitate multistep proton-coupled electron transfer steps. Therefore, the spin and magnetic-related principle has provided a promising pathway for engineering advanced ORR electrocatalysis, because the paramagnetic oxygen species are reduced to diamagnetic intermediates via spin-electron evolution during multiple electron transfer processes[17,18]. Recently, tuning the spin state of single-atom Fe centres has been achieved to develop advanced Fe-N-C catalysts. Therefore, this spin occupancy state modulation strategy in which the Pt $d$-orbital can regulate the adsorption state of $O_2$ and oxygen-relevant intermediates is a potential way to optimize ORR kinetics. However, to the best of our knowledge, tuning the spin occupancy state of Pt sites to accelerate the adsorption and dissociation kinetics of $O_2$ and oxygen-relevant intermediates for developing high-performance low-Pt catalysts remains largely unexplored. Herein, to boost the performance of low-Pt catalysts, we directed the charge injection to the Pt sites with an appropriate spin configuration and optimized the adsorption behavior via magnetic changes in the interactions of the dual-Fe sites from the PtFe alloy nanocrystals and atomically dispersed $FeN_4$ sites from a carbon substrate (denoted as PtFe@Fe$_{SAs}$-N-C). A combination of *operando* experiments and theories evidenced that the injected electrons of the Fe $dz^2$ orbital filled the perpendicular Pt $dz^2$ orbital and effectively achieved the side-on adsorption of $O_2$ and the dissociative pathway (direct 4e$^-$ pathway). Moreover, the σ* bond generated between the Pt $dz^2$ orbital and the 2p orbital of OH* accelerated the desorption of OH* and effectively inhibited the site-blocking effect, thereby facilitating the ORR kinetics. This process not only achieved electrochemical accessibility, but also restricted catalyst agglomeration and retarded the corrosion of carbon supports, metal detachment, and Ostwald ripening processes. Therefore, PtFe@Fe$_{SAs}$-N-C demonstrated a mass activity of 0.75 A mg$_{Pt}^{-1}$ at 0.9 V and a peak power density of 1240 mW cm$^{-2}$ in the full-cell assessment, outperforming commercial Pt/C catalysts (JM 20 wt%, 0.16 A mg$_{Pt}^{-1}$ and 1320 mW cm$^{-2}$), and a mass activity retention of 97% with no noticeable current drop at 0.6 V for over 220 h was attained; this mass activity retention exceeded the Department of Energy (DOE) technical targets for 2025.

## Results

### Synthesis and structural characterization

Recently, our group developed an adjacent ferromagnetic Mn (II) material that can adjust the electron spin state of the Fe (III) $e_g$ obital in N-coordinated dual-metal Mn,Fe-N$_6$-C, which is a high-activity ORR catalysts in acidic electrolytes[19]. Moreover, our previous data demonstrated the potential of the PtZn alloy nanocrystalline material as a low-Pt ORR catalyst[20], which, unfortunately, exhibited considerable activity loss at the rotating-disk electrode (RDE), thus, its performance under harsh fuel-cell conditions is questionable. We reasoned that the rational combination of a PtFe alloy with ferromagnetic microenvironments such as Fe-N-C materials could modulate the spin occupancy state of the Pt $dz^2$ orbital, thereby improving the 4e$^-$ ORR kinetics. Based on the above hypothesis, we fabricated PtFe intermetallic nanocrystals on a single atomically dispersed Fe-N-C substrate (Fe$_{SAs}$-N-C). Briefly, as shown in Fig. 1a and S1, the Fe$_{SAs}$-N-C substrate was first synthesized by annealing the dual-confined Fe, Zn-based zeolitic imidazolate framework-8 (Fe/ZIF-8) at 950 °C for 2 h in an argon flow. Then, the metal salts of H$_2$PtCl$_6$·6H$_2$O were dissolved in an isopropanol (reductant agent) solution mixed with Fe$_{SAs}$-N-C, followed by heating at 115 °C for 2 h, and the resulting powder was further annealed at 900 °C for 1 h in an argon flow to generate PtFe alloy nanocrystals on Fe$_{SAs}$-N-C (PtFe@Fe$_{SAs}$-N-C). Transmission electron microscopy (TEM) and high-angle annular dark-field scanning TEM

(HAADF-STEM) were employed to probe the fine structure of PtFe@Fe$_{SAs}$-N-C. Fig. S3 clearly demonstrates that Fe$_{SAs}$-N-C effectively inherits the initial dodecahedral shape of ZIF-8 without any discernible metallic species. Figure 1b,c and S5 clearly show PtFe alloy nanocrystals on PtFe@Fe$_{SAs}$-N-C with high density and excellent size uniformity of ~2.46 nm. As shown in Fig. S4, the evident decrease in the pore size distribution of PtFe@Fe$_{SAs}$-N-C at ~3 nm demonstrates the successful introduction of the PtFe alloy nanocrystals through microporous confinement and alloying effects (Table S1). Meanwhile, the retained rich surface area and pore structures ensure an expedited electron/mass transfer process. This process could also enhance the binding between the Pt alloy nanocrystals and the substrate, which prevents the alloy nanocrystals from aggregating. Powder X-ray diffraction (PXRD) shows (00 l), (111), and (200) peaks in the 2θ range of 23.96°, 41.04°, and 47.02°, respectively, indicating ordered stacking along the Z direction (Fig. S2). The wide-angle diffraction peak of PtFe@Fe$_{SAs}$-N-C at 41.04° can be perfectly indexed to the tetragonal phase of the PtFe structure with a lattice fringe of 3.71 Å (JCPDS 43-1359).

Figure 1c also shows that numerous isolated Fe atoms (labelled with red circles) anchored on the carbon substrate are present[21]. The identification of the isolated spots can be further confirmed by local electron energy loss spectroscopy (EELS) analysis (Fig. 1d)[7]. The HAADF-STEM images of the PtFe alloy nanocrystals (Fig. 1e and S6) indicate that the ordered intermetallic alloy nanocrystals are uniformly dispersed on Fe$_{SAs}$-N-C support. As shown in Fig. 1f-h, the characterized alternating bright and dim atomic columns as well as lattice distances of 0.187 nm (002) and 0.377 nm (001) reveal a standard ordered structure of the PtFe alloy, this results is, consistent with the atomic model, the HAADF-STEM image of individual alloy nanocrystals, and the XRD patterns of a tetragonal PtFe structure[22]. This deduction is further supported by the line intensity profile (Fig. 1i and S7). As shown in Fig. 1j, the energy-dispersive X-ray spectroscopy (EDX) mapping of PtFe@Fe$_{SAs}$-N-C obviously show the presence of Pt, Fe, N, and C. The microscopic characterization shows ordered alloy PtFe nanocrystals; these nanocrystals were successfully formed and anchored on the Fe$_{SAs}$-N-C substrate. According to the inductively coupled plasma (ICP) mass spectrometry results, the Fe and Pt loadings in PtFe@Fe$_{SAs}$-N-C are 7.75 and 6.97 wt%, respectively (Table S2).

### Electronic states and coordination environment of PtFe@Fe$_{SAs}$-N-C catalyst

We studied the electron transfer behavior of Pt and Fe in the formation of PtFe@Fe$_{SAs}$-N-C catalysts. Fig. S8 shows the wide scan X-ray photoelectron spectroscopy (XPS) results for the samples, and Table S3 summarizes the corresponding elemental compositions, these results, confirm the presence of Pt, Fe, C, and N. The deconvolution of the Pt 4$f$ spectra reveals the main peaks for Pt$^0$ along with Pt$^{2+}$ (Fig. S9a). The electron transfer from Pt to the Fe$_{SAs}$-N-C support causes a positive shift in the Pt 4$f$ binding energy in PtFe@Fe$_{SAs}$-N-C compared to Pt/C[23]. The FeN$_4$ sites embedded in the carbon substrate, which shows an electronegative behaviour,, can modify the electronic structure of adjacent carbon. The resultant electron deficiency of carbon strengthens the deposition of Pt and the metal-support interaction[24]. The deconvoluted N 1$s$ spectra of Fe$_{SAs}$-N-C and PtFe@Fe$_{SAs}$-N-C (Fig. S9b) include a dominant graphitic N along with pyridinic N, metal N, and oxidized N[23,25]. The existence of N-Fe peaks confirms the coordination between Fe and N in the carbon support. When Fe atoms are alloyed with Pt, the N-Fe bonds are broken, resulting in a significant decrease in the content of N-Fe and pyridinic N (Table S4). The N-Fe bonds and pyridinic N generally evidence the formation of FeN$_4$ active sites (Fig. S10), thus, the FeN$_4$ active site is effectively preserved for all the corresponding PtFe@Fe$_{SAs}$-N-C catalysts, which is consistent with the HAADF-STEM image analysis results.

X-ray absorption structure (XAS) spectra were recorded to elucidate the electronic structures of PtFe@Fe$_{SAs}$-N-C. No evident

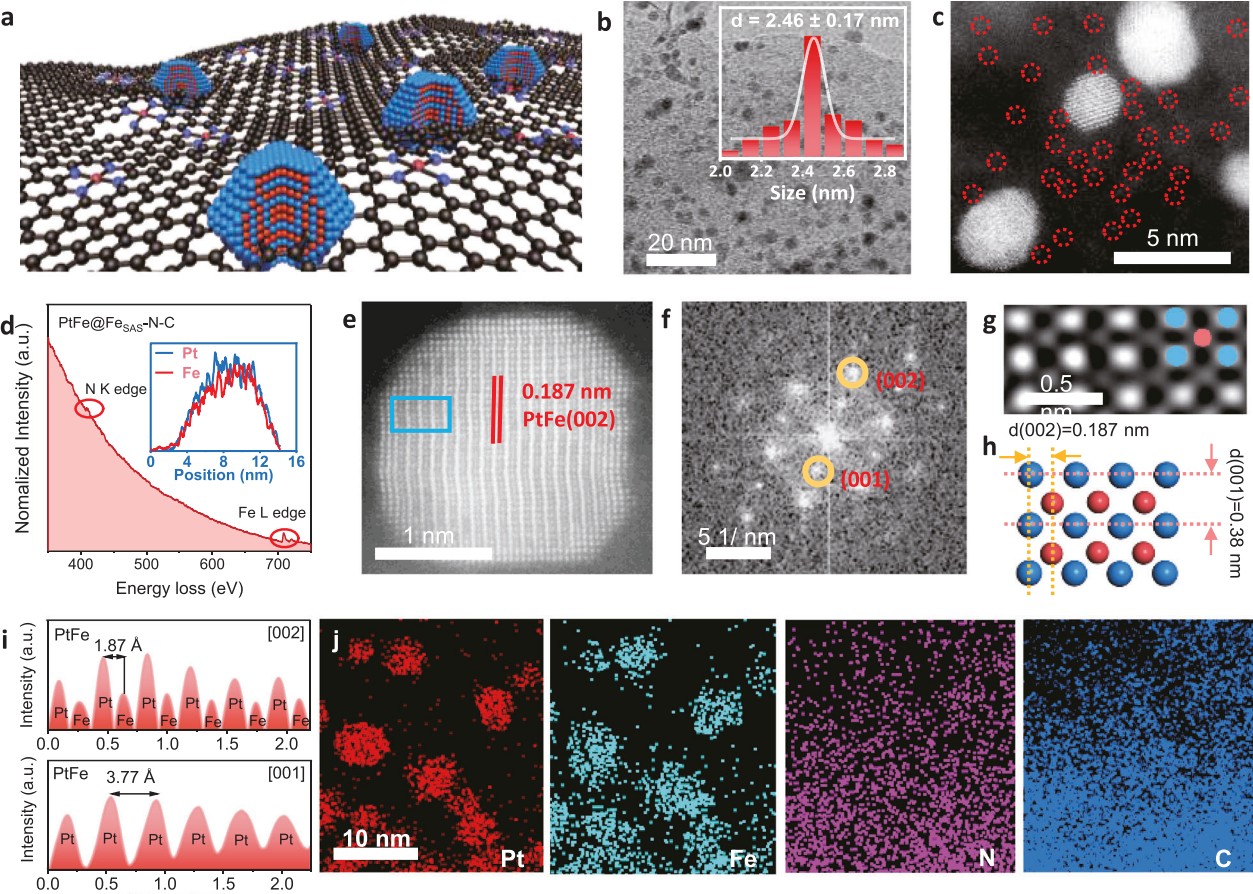

**Fig. 1 | Structure characterizations. a** Scheme of PtFe@Fe$_{SAs}$-N-C. **b** TEM images of the PtFe@Fe$_{SAs}$-N-C showing a uniform distribution of Pt nanocrystals. **c** HAADF-STEM image and **d** corresponding EELS analysis to verify the coexistence of PtFe and atomic level Fe in the PtFe@Fe$_{SAs}$-N-C electrocatalyst (the spots in the red dashed circles is ascribed to the Fe single atoms, inset: differences in mass content between Fe and Pt in PtFe alloy nanoparticles.). **e** HAADF-STEM image of an individual PtFe nanocrystals. **f** FFT pattern of Fig. 1e. **g** The enlarged HAADF-STEM image in the blue box in **e**. **h** Crystal structure of PtFe with ordered, tetragonal structure and its projection along the (001) zone axis (blue and red spheres represent Pt and Fe, respectively). **i** Intensity profile across the PtFe particle from (002) and (001) directions measured from h. The distance between the strong intensities matches with the separation between Pt atoms. **j** EDS mapping results of PtFe@Fe$_{SAs}$-N-C.

differences were observed in the Pt L$_3$-edge X-ray absorption near-edge structure (XANES) spectra (Fig. 2a) in the pre-edge region compared with that of Pt foil. The stronger intensity of the white line for PtFe@Fe$_{SAs}$-N-C results from the electron transfer from Pt 5$d$ to the carbon support, and further demonstrates their enhanced metal-support interaction[20,26]. The negative shift of the main peak from 2.51 Å (Pt foil) to 2.36 Å (PtFe@Fe$_{SAs}$-N-C) in the Fourier transforms of the extended XAFS (FT-EXAFS) data (Fig. S11) affirms the formation of Pt-Fe bonds. The Fe K-edge XANES spectra in Fig. 2b, the pre-edge of PtFe@Fe$_{SAs}$-N-C shows a shift towards lower energy compared to that of Fe$_{SAs}$-N-C and FePc, these results indicate the reduction of Fe$^{3+}$ in the carbon matrix during the alloying process. The linear relationship between the valence of Fe and the pre-edge location shows that the valence of Fe in PtFe@Fe$_{SAs}$-N-C is approximately +0.92 (Fig. 2c). Moreover, the FT-EXAFS spectra of the R space show that the shoulder at 2.48 Å originates from the scattering by the Fe-Pt bond, whereas the strong Fourier transform peak at 1.53 Å is attributed to scattering from the Fe-N bond in the carbon support (Fig. 2d)[27]. The coordination structures of Pt and Fe are also further supported by wavelet transform EXAFS analysis (Fig. S12). Ultraviolet photoelectron spectroscopy (UPS) (Fig. 2e and S13) was performed to obtain a better understanding of the band structure. Intriguingly, the cut-off energies ($E_{cutoff}$) of PtFe@Fe$_{SAs}$-N-C and Fe$_{SAs}$-N-C are 17.59 and 16.97 eV, respectively.

According to the equation of e$_\Phi$= 21.22 eV-$E_{cutoff}$[17], the work functions (e$_\Phi$) of PtFe@Fe$_{SAs}$-N-C and Fe$_{SAs}$-N-C were calculated to be 3.63 and 4.25 eV, respectively; these results indicate that PtFe@Fe$_{SAs}$-N-C tends to donate more electrons to the O$_2$ molecules/oxygen-relevant intermediates[28]. Furthermore, the energies of the highest occupied molecular orbital (HOMO) were determined to be 2.65 and 2.24 eV for PtFe@Fe$_{SAs}$-N-C and Fe$_{SAs}$-N-C catalysts, respectively, indicating that electron delocalization occurred in the Fe sites[17,25]. As the charge redistribution is accompanied by the transition of the 3$d$ electron spin configuration, an electron spin resonance (ESR) test was employed to determine the spin states of the Fe in PtFe@Fe$_{SAs}$-N-C. As the curves displayed in Fig. 2f, the major characteristic signals at $g$ = 4.48 are attributed to the Fe(III) medium-spin states (S = 3/2), and the minor signals at $g$ = 2.12 and 2.09 are assigned to the Fe(III) high-spin states (S = 5/2)[29,30]. Meanwhile, the Fe(III) signal in the ESR spectra of PtFe@Fe$_{SAs}$-N-C decreases relative to Fe$_{SAs}$-N-C, which may be due to partial alloying of the Fe atoms. As a solid foundation, the electron-spin state of Fe in PtFe@Fe$_{SAs}$-N-C was measured by $^{57}$Fe Mössbauer spectroscopy (Figs. S14,15 and Table S5). Bartholomew and Boudart suggested that the Mössbauer absorption probability of the surface Fe atoms in an alloy is substantially the same as that in the bulk, especially for nanosized PtFe particles[31,32]. Thus, PtFe@Fe$_{SAs}$-N-C exhibited a dominant resonant line sextet and iron metal component, which was

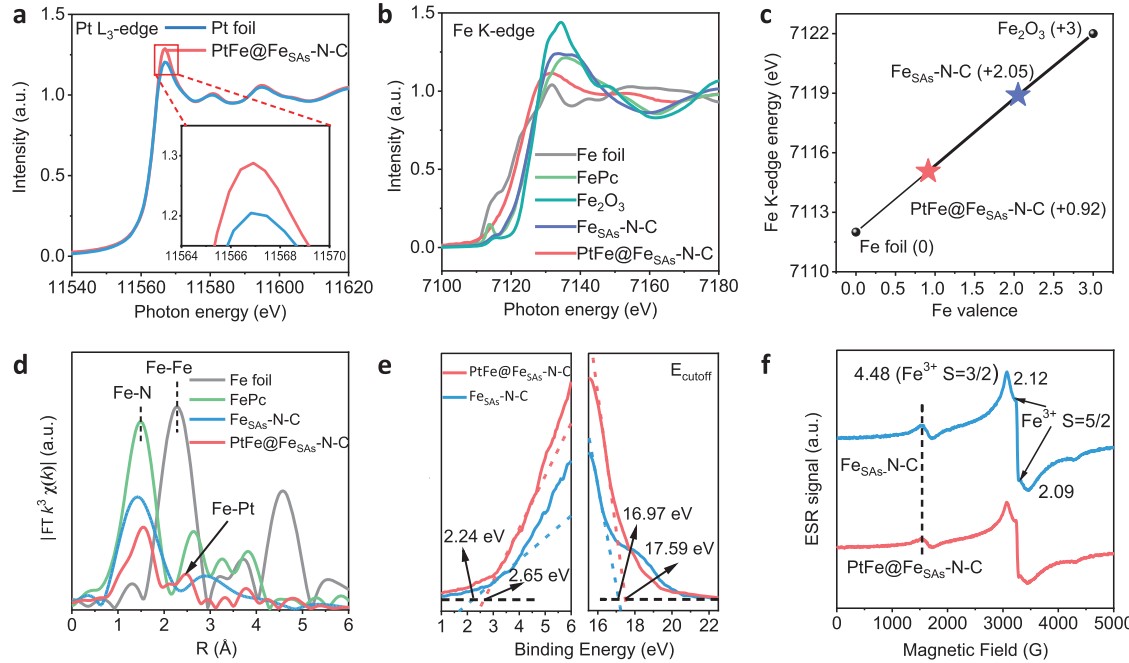

**Fig. 2 | Spectroscopy analysis.** Comparisons of the X-ray absorption near-edge structure spectra of PtFe@Fe$_{SAs}$-N-C, Fe$_{SAs}$-N-C, Fe foil, FePc, and Pt foil: **a** XANES spectra of PtFe@Fe$_{SAs}$-N-C and Pt foil: Pt L$_3$ edge. **b** XANES spectra of PtFe@Fe$_{SAs}$-N-C, Fe$_{SAs}$-N-C, Fe foil, and FePc: Fe K edge. **c** Liner fitting for Fe valences in PtFe@Fe$_{SAs}$-N-C, Fe$_{Sas}$-N-C, Fe$_2$O$_3$, and Fe foil derived from corresponding Fe K-edge XANES spectra. **d** FT-EXAFS spectra of PtFe@Fe$_{SAs}$-N-C, Fe$_{Sas}$-N-C, Fe foil, and FePc at the Fe K-edge. **e** Enlarged UPS spectra. **f** X-band EPR spectra of PtFe@Fe$_{SAs}$-N-C and Fe$_{SAs}$-N-C.

attributed to the formation of the PtFe alloy, and the spectral area of PtFe reached 81.5%. Besides, three quadrupole split doublets were observed at the center of the center shift: 0.84 mm s$^{-1}$, 2.19 mm s$^{-1}$, and 3.20 mm s$^{-1}$ were ascribed to medium-spin Fe(III)N$_4$ (11.5 %), high-spin Fe(III)N (2.1 %), and medium-spin Fe(II)N$_4$ (4.9 %) in the Fe$_{SAs}$-N-C substrate, respectively. To monitor the evolution of the magnetic properties of PtFe@Fe$_{SAs}$-N-C as a function of Fe loading, magnetization measurements were carried out[33]. According to the saturation magnetizations (Fig. S16), the average magnetic moments of the individual Fe atom in Fe$_{SAs}$-N-C and PtFe@Fe$_{SAs}$-N-C were determined to be 4.96 and 4.37 $\mu_B$, respectively. The decrease in the magnetic moment of the Fe atoms with low Pt loading was consistent with the ESR results; this result indicated a redistribution of the electronic structure in the catalyst, which was beneficial for the attainment of the excellent catalytic performance of PtFe@Fe$_{SAs}$-N-C.

## ORR performance

The ORR electrocatalytic activity of the catalysts was evaluated by a rotating-ring disk electrode (RRDE) in an O$_2$-saturated 0.1 M HClO$_4$ electrolyte at room temperature, without *iR* correction. As shown in Fig. 3a and S17, the ORR activities were measured by linear sweep voltammetry (LSV) at a rotation speed of 1600 rpm. Obviously, PtFe@Fe$_{SAs}$-N-C with multiple active sites exhibits the highest ORR activity, with a half-wave potential (E$_{1/2}$) of 0.872 V *vs*. RHE, displaying positive shifts of 102 and 43 mV relative to Fe$_{SAs}$-N-C and commercial Pt/C catalysts, respectively. Furthermore, the Koutecky-Levich (K-L) plots (Fig. S18) and the RRDE results (Fig. S19) confirm the 4e$^-$ transfer process for PtFe@Fe$_{SAs}$-N-C (~ 3.5%) with a lower H$_2$O$_2$ yield than that of Fe$_{SAs}$-N-C (~10%). As shown in Fig. S20, PtFe@Fe$_{SAs}$-N-C exhibits a kinetic current density of 25.82 mA cm$^{-2}$ at 0.85 V *vs*. RHE, this density is greatly improved with respect to Fe$_{SAs}$-N-C (2.15 mA cm$^{-2}$) and Pt/C (11 mA cm$^{-2}$). The mass activity and specific activity (SA) of PtFe@Fe$_{SAs}$-N-C are 1.18 A mg$_{Pt}^{-1}$ and 3.64 mA cm$^{-2}$, respectively, at 0.85 V *vs*. RHE, these are superior to those of commercial Pt/C (0.21 A mg$_{Pt}^{-1}$, 334 mA cm$^{-2}$) catalysts (Figs. S21 and S22). As shown in Fig. S23,

PtFe@Fe$_{SAs}$-N-C (11.2 Ω) shows lower resistance than both Fe$_{SAs}$-N-C (17.1 Ω) and Pt/C (18.2 Ω), indicating the effective promotion of rapid electron/proton transfer. Specifically, to understand the origin of the excellent ORR performance of PtFe@Fe$_{SAs}$-N-C, anionic SCN$^-$ (10 mM) poisoning measurements were carried out in 0.1 M HClO$_4$ electrolyte (Fig. 3c). SCN$^-$ has a strong affinity for Fe ions and is used as a probe to poison the Fe-N$_4$ sites[34]. As a result, the slight decrease of E$_{1/2}$ reveals that the single-atom FeN$_4$ site around the alloy nanocrystals also exhibits certain activity; however, E$_{1/2}$ is still greater than 0.81 V *vs*. RHE, indicating that the PtFe alloy nanocrystals are responsible for the high ORR activity of PtFe@Fe$_{SAs}$-N-C. The Fe atoms in the support mainly provide Fe sources for alloying and anchoring PtFe nanocrystals to inhibit agglomeration and demetallization.

Next, we assessed the long-term stability of PtFe@Fe$_{SAs}$-N-C in an O$_2$-saturated 0.1 M HClO$_4$ electrolyte. The E$_{1/2}$ of PtFe@Fe$_{SAs}$-N-C (Fig. 3a) decreases by only 9 mV after 50,000 potential cycles between 0.6 and 1.0 V *vs*. RHE potential windows, this is, much lower than those of Fe$_{SAs}$-N-C (71 mV) and Pt/C (88 mV) (Fig. S24). Accordingly, the retained mass activity of PtFe@Fe$_{SAs}$-N-C is 1.031 A mg$_{Pt}^{-1}$ and 0.928 A mg$_{Pt}^{-1}$ after 30,000 and 50,000 potential cycles, respectively; specifically, the losses are 12.5% and 9.9% respectively, in the mass activity, and these losses are much lower than the 70.7% and 48.5%, respectively, of Pt/C (Fig. 3b). To evaluate the stability of the catalysts at a more practical temperature, the temperature of the testing system was increased to 60 °C. Notably, after 30,000 potential cycles, PtFe@Fe$_{SAs}$-N-C still demonstrates high stability, with a minimal E$_{1/2}$ loss of only 28 mV (Fig. S25) and 70% mass activity retention, while the Pt/C catalyst suffers more severe degradation of ORR activity (Fig. S26), with only 17% mass activity retention (inset in Fig. S25a). All the above results verify that PtFe@Fe$_{SAs}$-N-C has greater activity and greater stability than Fe$_{SAs}$-N-C, Pt/C, and previous reports of Pt-based electrocatalysts (Fig. S25b).

One of the major concerns with M-N-C catalysts in fuel cells is the formation of a large amount of reactive oxygen species (ROS, e.g. H$_2$O$_2$) as a byproduct, which is detrimental to the membrane and

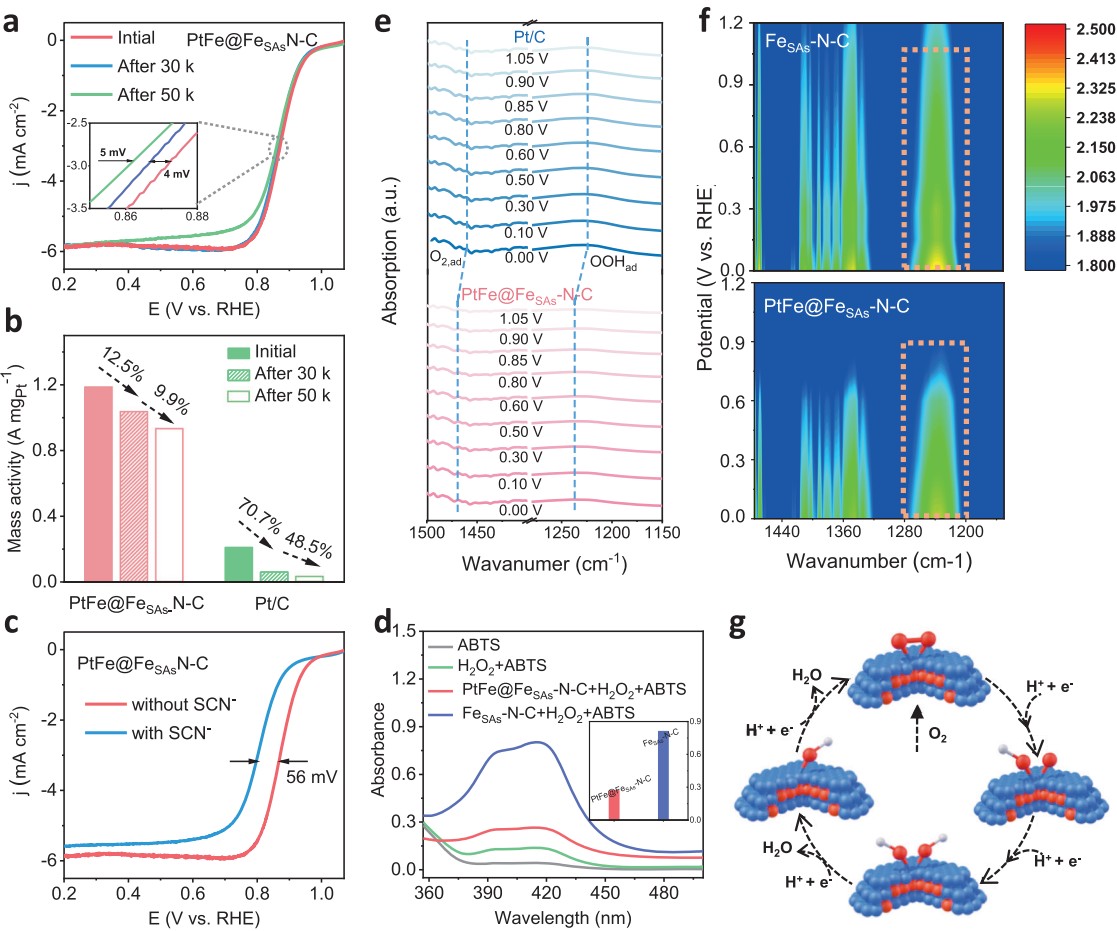

**Fig. 3 | ORR performances, in-situ FTIR spectroscopy, and reaction pathway analysis. a** Linear sweep voltammetry polarization curves under a rotation rate of 1600 rpm for PtFe@Fe$_{SAs}$-N-C after 30 k and 50 k potential cycles (0.6-1.0 V vs. RHE) in O$_2$-saturated 0.1 M HClO$_4$ electrolyte at 25 °C (without iR compensation, catalyst loading: 0.1 mg cm$^{-2}$). **b** The mass activity loss ratio of PtFe@Fe$_{SAs}$-N-C and Pt/C at 0.85 V after 30 k and 50 k potential cycles. **c** Linear sweep voltammetry polarization curves under a rotation rate of 1,600 rpm for PtFe@Fe$_{SAs}$-N-C in O$_2$-saturated 0.1 M HClO$_4$ electrolyte at 25 °C before and after the addition of 10 mM SCN$^-$ (without iR compensation). **d** UV-Vis absorption spectra of 0.1 M HClO$_4$ electrolyte of ABTS + H$_2$O$_2$ with PtFe@Fe$_{SAs}$-N-C, Fe$_{SAs}$-N-C catalyst and no catalyst. Inset: the absorbance value of the solution at 417 nm. **e, f** The comparison of in-situ FTIR spectra under applied potentials among PtFe@Fe$_{SAs}$-N-C, Pt/C, and Fe$_{SAs}$-N-C. **g** Proposed ORR reaction pathways occurring on the Pt site in PtFe@Fe$_{SAs}$-N-C (blue and red spheres represent Pt and Fe, respectively).

ionomers[35,36]. As reported, 2,20-azinobis(3-ethylbenzthiazoline-6-sulfonate) (ABTS) can be oxidized by ROS, resulting in a change in absorbance at ~417 nm in UV-Vis absorption spectroscopy, this, is mainly manifested as the solution changes from colourless to green (Fig. 3d and S27)[37]. As expected, the UV-Vis absorbance intensity of Fe$_{SAs}$-N-C + H$_2$O$_2$ + ABTS is approximately 3 times that of PtFe@Fe$_{SAs}$-N-C + H$_2$O$_2$ + ABTS, thus, the introduction of Pt can significantly inhibit the Fenton reaction and the accumulation of ROS. This deduction is consistent with previous studies, showing mixing Pt[38] and/or Pt-Co alloy[6] particles as peroxide/radical scavengers with M-N-C can alleviate H$_2$O$_2$ accumulation.

The morphology and structure of PtFe@Fe$_{SAs}$-N-C after the stability tests in the RRDE were further analyzed by HAADF-STEM and STEM-EELS. As shown in S28, after 50,000 potential cycles, the sizes of the PtFe nanocrystals are effectively preserved at 3.35 nm without apparent agglomeration. These results support our hypothesis that the existence of the FeN$_4$ sites in the supports strengthens the metal-support interaction, leading to less detachment and agglomeration of the PtFe nanocrystals. Also, abundant single atoms are still uniformly distributed in the carbon support (Fig. S29a). The lattice spacing and ordered structure characterized in Fig. S29b are consistent with those of the PtFe nanocrystals in the catalyst before potential cycling (Fig. 1e), because of the ordered tetragonal crystal structure. The EELS

analysis (Fig. S29c) verified the preservation of single Fe atoms with an N-coordinated configuration, similar to that of the pristine sample (Fig. 1d). Significantly, the PtFe nanocrystals on Fe$_{SAs}$-N-C support maintain their intermetallic compound structures due to the strong metal-support interaction (Fig. S29d), which plays an important role in maintaining the stability of PtFe@Fe$_{SAs}$-N-C. To better demonstrate that the strong metal-support interaction and the formation of ordered PtFe alloys can effectively suppress the leaching of Pt and Fe metals, XPS characterization was used to measure the retention rates of metal Pt and Fe metals in the catalyst after stability testing. As shown in Fig. S30, after 30,000 potential cycles, for Fe$_{SAs}$-N-C, the amount of leached Fe is as high as 45.1%, while it is greatly reduced to 14.2% for PtFe@Fe$_{SAs}$-N-C. For Pt/C, the amount of leached Pt is as high as 63%, whereas it is only 12.5% for PtFe@Fe$_{SAs}$-N-C. Furthermore, the excellent stability of the PtFe structure was confirmed by comparing the ICP-MS, Raman, and XRD results of the catalyst before and after potential cycling (Figs. S31,32 and Table S6).

## Catalytic mechanism analysis

In-situ FTIR, *operando* XAFS, and *operando* magnetometry techniques were employed to elucidate the catalytic mechanism of the multistep ORR based on spintronic behavior and bond order theory. As shown in Fig. 3e,f and S33, Pt/C and Fe$_{SAs}$-N-C exhibit a readily identifiable OOH*

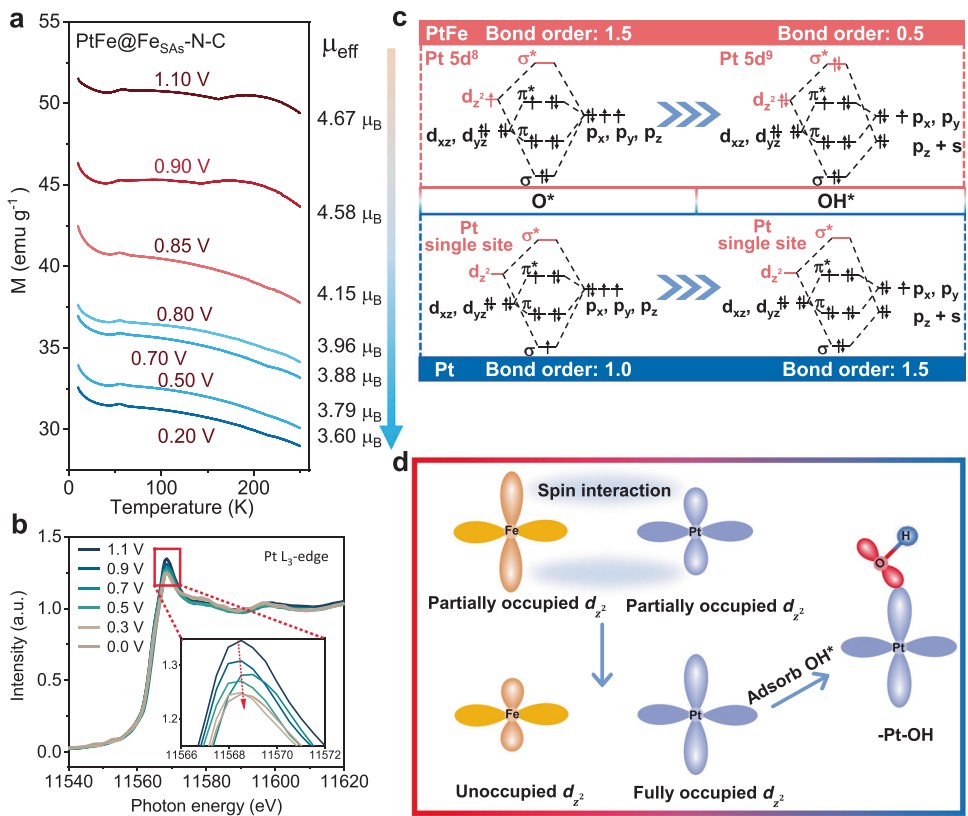

**Fig. 4 | *Operando* magnetometry, *operando* XAFS measurements, and catalytic mechanism analysis. a** The M-T curve and effective magnetic ($\mu_{eff}$) moment of the PtFe@Fe$_{SAs}$-N-C vary with the applied potential in the O$_2$-saturated 0.1 M HClO$_4$ electrolyte. **b** *Operando* Pt L$_3$-edge XANES spectra for PtFe@Fe$_{SAs}$-N-C in 0.1 M HClO$_4$. **c** The orbital interactions between Pt site and the ORR (O* and OH*) intermediates on PtFe@Fe$_{SAs}$-N-C and Pt single site. **d** A schematic diagram of Pt-Fe spin charge injection effect for O$_2$ activation.

peaks at 1226 and 1234 cm$^{-1}$, and strong O-O stretching modes of O$_{2,ad}$ at 1462 and 1469 cm$^{-1}$, respectively, with increasing intensity caused by decreasing the applied potentials; this shows an associative pathway of Pt/C and Fe$_{SAs}$-N-C[39]. However, the OOH* peak of PtFe@Fe$_{SAs}$-N-C (1237 cm$^{-1}$) is significantly redshifted and attenuated compared with that of Pt/C, indicating that the 4e$^-$ reaction pathway changed from the indirect pathway (association pathway) of the Pt/C catalyst to the direct pathway (dissociative pathway, Griffiths adsorption model). As shown in Fig. 3g, this path does not generate OOH* and can strongly inhibit the production of byproducts, which is beneficial for improving the ORR stability.

The real-time magnetic responses accompanying the electrochemically-driven reactions were recorded to investigate the magnetization evolution during the ORR process. As shown in Fig. 4a and S34, the magnetization and effective magnetic moment of PtFe@Fe$_{SAs}$-N-C gradually decrease as the applied potential decreases from 1.1 to 0.2 V *vs.* RHE; this decrease is attributed to the Griffiths adsorption model of O$_2$ molecules and the optimization of O$_2$ molecule/intermediate adsorption behavior on the active Pt site[40]. Kelvin probe force microscopy (KPFM) was used to probe the variations in e$_\Phi$ between PtFe nanocrystals and Fe$_{SAs}$-N-C[41]. KPFM was conducted with a derivative imaging mode of atomic force microscopy to monitor the changes in the local e$_\Phi$. The calibration standard sample was highly oriented pyrolytic graphite (HOPG, e$_\Phi$ = 4.6 eV). As shown in Fig. S35, the topographical image and contact potential difference (CPD) image of PtFe@Fe$_{SAs}$-N-C were simultaneously recorded, and the corresponding acquired profiles were also obtained. The CPD of the PtFe nanocrystals is approximately −100 mV (e$_\Phi$ = 4.8 eV) lower than that of the probe, while that of Fe$_{SAs}$-N-C is 300 mV (e$_\Phi$ = 4.4 eV) greater than that of the probe, implying that electrons tend to flow from the Fe$_{SAs}$-

N-C support to the PtFe nanocrystals. Moreover, due to the electronegativity difference between Pt ($\chi$ = 2.28) and Fe ($\chi$ = 1.83) metal atoms, electrons have a driving force to transfer from Fe to Pt. Specifically, this is further confirmed by *operando* XANES of the Pt L$_3$-edge and Fe K-edge. In *operando* XANES of the Pt L$_3$-edge (Fig. 4b), with the ORR process, and given the applied potential decreasing from 1.1 to 0.0 V, the white line intensity continuously decreases, revealing an increase in the electron occupied state of the Pt 5d orbital. As expected, *operando* XANES of the Fe K-edge shows the opposite phenomenon (Fig. S36). When the applied potential decreases from 1.1 to 0.0 V *vs.* RHE, the absorption edge continuously shifted towards high energies; thus, Fe in PtFe@Fe$_{SAs}$-N-C lost electrons and the valence state significantly increased during the ORR process. The valence state fitting of ex-situ and *operando* Pt L$_3$-edge XANES also confirmed this result (Fig. S37). To our best knowledge, the strong bonding of OH* over the Pt sites leads to sluggish ORR kinetics, which is a common challenge for Pt-based catalysts[16]. Therefore, the orbital interactions between the Pt sites and OH* were then investigated. As reported, in 0.1 M HClO$_4$ media, the ORR kinetically slow O* → OH* reduction peak is located between 0.55 and 1.0 V *vs.* RHE[42]. As shown in Fig. 4a, b, S36, and 37, when the potential decreases from 1.1 to 0.5 V *vs.* RHE, the intensity of the white line representing the Pt L$_3$-edge decreases, the absorption edge of the Fe K-edge positively shifts and the effective magnetic moment significantly decreases, which strongly confirms prove that the number of Pt 5d electrons occupied increases when they interact with OH*.

The bond order theorem was applied to describe the potential orbital interactions between the adsorption intermediates and Pt sites at the electrocatalyst surface. According to the principle of symmetry conservation, we omitted the orbitals ($dx^2$-$y^2$ and $dxy$) of Pt as shown in

Fig. 4c, d. Since the orbital interaction strength between Pt and the oxygen-containing intermediates is proportional to the bond order, we calculated the bond order of the O* and OH* intermediates bound to Pt in PtFe@Fe$_{SAs}$-N-C using the bond order = (A-B)/2 formula, where A and B are bonding electrons and antibonding electrons, respectively. Underorbital interactions, the bond order between Pt sites and adsorbed state O* is 1.5, which is 1 greater than the bond order between Pt and OH*, indicating that the adsorption of O* by Pt sites is significantly stronger than that of OH*. From a thermodynamic perspective, this is conducive to the desorption of the products. Notably, this phenomenon is opposite at the Pt single site[16]. This occurs because the production of an additional σ* bond between the fully occupied Pt $dz^2$ orbital of PtFe@Fe$_{SAs}$-N-C and the π orbital of the OH* intermediates, results in a weakened Pt-OH* bonding strength; this is beneficial for suppressing the site-blocking effect, and effectively facilitating the ORR kinetics. Moreover, O$_2$ temperature-programmed desorption (O$_2$-TPD, Fig. S38) measurements were also performed to determine the influence of the PtFe nanocrystals on the catalytic performance. The Pt-O peak of PtFe@Fe$_{SAs}$-N-C shows a desorption temperature comparable to that of Fe-O of Fe$_{SAs}$-N-C, but with a stronger intensity of the O$_2$ desorption peak, which implies that the Pt site in PtFe@Fe$_{SAs}$-N-C has a strong adsorption affinity for O$_2$, which is favourable for the activation of the O-O bond. Consequently, the above results powerfully confirm that the spin charge injection effect between Pt and Fe can achieve the Griffiths model ORR pathway on PtFe@Fe$_{SAs}$-N-C, preventing the site-blocking effect and formation of H$_2$O$_2$ to enhance both selectivity and stability.

Density functional theory (DFT) calculations were performed to further explore the origin of the high activity and durability of PtFe@Fe$_{SAs}$-N-C. Based on the experimental results, the optimization of the catalytic reaction behavior at the Pt site is derived from the bidirectional regulation of Fe atoms in the substrate and alloy; thus, PtFe(111), Pt(111)/FeN$_4$, Pt(111), and FeN$_4$, simulation models were constructed to represent the possible types of active sites (Fig. S39 and supplementary data 1). Periodic stacked magnetic multilayer structures consisting of two magnetic layers separated by a nonmagnetic spacer (i.e., Pt), are known to exhibit special interlayer exchange coupling interactions[43]. In this study, the hybridized d-orbitals of Pt and Fe constitute the bypass transfer of magnetization between the different layers throughout the entire structure[44]. This Pt-Fe hybrid involving 3d-5d orbitals can be clearly seen in the density of states (DOS) of PtFe(111) structure (Fig. S40). Furthermore, the spin density distribution of the PtFe(111) structures was calculated to confirm the polarization of Pt atoms by adjacent Fe atoms, with an average magnetic moment of 0.371μ$_B$ (Fig. S41). To verify the intrinsic driving mechanism of the electron structure of the Pt site regulated by Fe atoms in the substrate, we calculated the e$_\Phi$ of the PtFe(111) and FeN$_4$ structural models. The energy level of e$_\Phi$ determines the direction and possibility of electron transfer, and the electrons can more easily leave the metal when the e$_\Phi$ value is lower.. As shown in Fig. 5a and S42, the e$_\Phi$ of FeN$_4$ (4.02 eV) is significantly lower than that of PtFe(111) (4.93 eV) and consistent with KPFM measurements. Therefore, in PtFe@Fe$_{SAs}$-N-C, a built-in electric field is formed at the interface of the PtFe cluster and Fe$_{SAs}$-N-C, and electrons act as a driving force from Fe$_{SAs}$-N-C with low e$_\Phi$ (4.02 eV) to the PtFe cluster with high e$_\Phi$ (4.93 eV)[41], thus, this process jointly regulates the adsorption and desorption behavior of Pt sites towards O$_2$ and oxygen-relevant intermediates with Fe atoms on the surface of clusters.

To evaluate whether the ORR dissociative pathway occurs in various simulation models, the key intermediates of O + OH*, 2OH*, and OH* (*denotes the adsorbed state) were optimized. The determined potential energy profiles show that PtFe(111) (0.54 eV) and Pt(111)/FeN$_4$ (0.56 eV) possess a lower overpotential than Pt(111) (1.11 eV) and FeN$_4$ (0.74 eV), which confirmed the better catalytic performance of PtFe@Fe$_{SAs}$-N-C (Fig. 5b). Notably, the formation and

dissociation of O + OH* intermediates efficiently result in more favourable direct 4e$^-$ processes (Fig. S43). However, these two processes are not easily performed on the FeN$_4$ and Pt(111) surfaces because O + OH* cannot spontaneously form according to thermodynamics; its formation, would greatly affect the catalytic reaction process and selectivity. Corresponding to the potential of activity evaluation in the experiment, the Gibbs free energy diagrams of the ORR on two possible active sites were constructed at 0.85 V. As shown in Fig. 5c-e, PtFe(111) and Pt(111)/FeN$_4$ exhibit excellent ORR activity, both with rate-determining steps (RDS) at OH* protonation, and reaction energy barriers of only 0.16 and 0.18 eV; these barriers are much smaller than those for Pt(111) and FeN$_4$ (RDS are the second OH* protonation, with energy barriers of 0.73 and 0.32 eV, respectively), and they alleviate the site-blocking effect of OH*, to facilitating the ORR kinetics. Based on the above results, we constructed a PtFe(111)/FeN$_4$ structural model to theoretically verify the experimental results. First, the adsorption energy of PtFe(111) on the FeN$_4$ support was calculated, and the results showed that the adsorption energy was −0.792 eV, which confirmed the stability of the PtFe(111)/FeN$_4$ structure. Second, the charge density difference of the PtFe(111)/FeN$_4$ structure was calculated, as shown in Fig. S44. Clearly, charge aggregation occurs between PtFe(111) and the FeN$_4$ support, which suggests a strong metal-support interaction between the PtFe(111) nanoparticles and the FeN$_4$ support. The step diagram of the 4e$^-$ ORR carried out on the surface of the PtFe(111)/FeN$_4$ model was further calculated, as shown in Fig. S45. For the PtFe(111)/FeN$_4$ structure, the RDS is also the first OH*protonation, as experimentally confirmed, its RDS energy barrier is the lowest of all structures at only 0.033 eV, confirming that the PtFe(111)/FeN$_4$ structure is more favourable for the 4e$^-$ ORR.

Therefore, the strong metal-support interactions and spin-charge injection effects between Pt and Fe atoms are beneficial for ORR enhancing activity and stability by optimizing O$_2$ molecule activation/dissociation and the OH* desorption process. For single-atom FeN$_4$ sites, the inappropriate adsorption of oxygen-related intermediates (O$_2$*, OH*, and OOH*) on the active site can easily lead to active site poisoning and corrosion problems, resulting in sluggish kinetics(Fig. S46)[13,45]. Compared to Pt(111) and FeN$_4$ sites, the PtFe(111)/FeN$_4$ structure exhibits a unique ORR catalytic mechanism, with the optimal adsorption state of oxygen intermediates and the ability to inhibit the occurrence of side reactions. Combined with the low RDS, the OH* protonation energy barrier of PtFe(111)/FeN$_4$ can effectively improve the ORR kinetics of PtFe@Fe$_{SAs}$-N-C and increase its stability.

## Applications in the PEMFC system

PtFe@Fe$_{SAs}$-N-C, Fe$_{SAs}$-N-C, and commercial Pt/C catalysts were equipped into the membrane electrode assembly (MEA) as a cathode to evaluate fuel cell performances under 28 psi H$_2$-O$_2$ environment (Fig. 6a, b and S47). Compared with Fe$_{SAs}$-N-C (0.54 W cm$^{-2}$) and typical Pt/C (1.38 W cm$^{-2}$, 0.2 mg$_{Pt}$ cm$^{-2}$), the fuel cell with PtFe@Fe$_{SAs}$-N-C shows significantly enhanced performance with a power density of 1.24 W cm$^{-2}$ (0.12 mg$_{Pt}$ cm$^{-2}$). The PtFe@Fe$_{SAs}$-N-C cell in a H$_2$-air environment (Fig. 6c) shows a better performance than that of Fe$_{SAs}$-N-C in the whole current density range with a higher peak power density (0.71 vs. 0.42 W cm$^{-2}$), this, also significantly outperformed the 2021 DOE milestone, and achieved current densities of 240 mA cm$^{-2}$ at 0.8 V (>60 mA cm$^{-2}$) and 690 mA cm$^{-2}$ at 0.675 V (≥300 mA cm$^{-2}$)[46]. The mass activity of PtFe@Fe$_{SAs}$-N-C calibrated to an absolute H$_2$ and O$_2$-air pressure of 28 psi from H$_2$/O$_2$ and H$_2$/air polarization curves is 0.75 and 0.70 A mg$_{Pt}$$^{-1}$ at 0.9 V$_{iR-free}$, respectively; these values are around 4.6 times greater than that of Pt/C (0.16 A mg$_{Pt}$$^{-1}$), and 1.7 times higher of the 2025 DOE activity target (0.44 A mg$_{Pt}$$^{-1}$) (Fig. 6e)[26]. The mass activity of PtFe@Fe$_{SAs}$-N-C is higher than those of most Pt-based and hybrid electrocatalysts (Table S8). In addition, PtFe@Fe$_{SAs}$-N-C cell was subjected to an accelerated durability test according to the DOE protocols[7]. As shown in Fig. 6d and S48, the i-v polarization curve

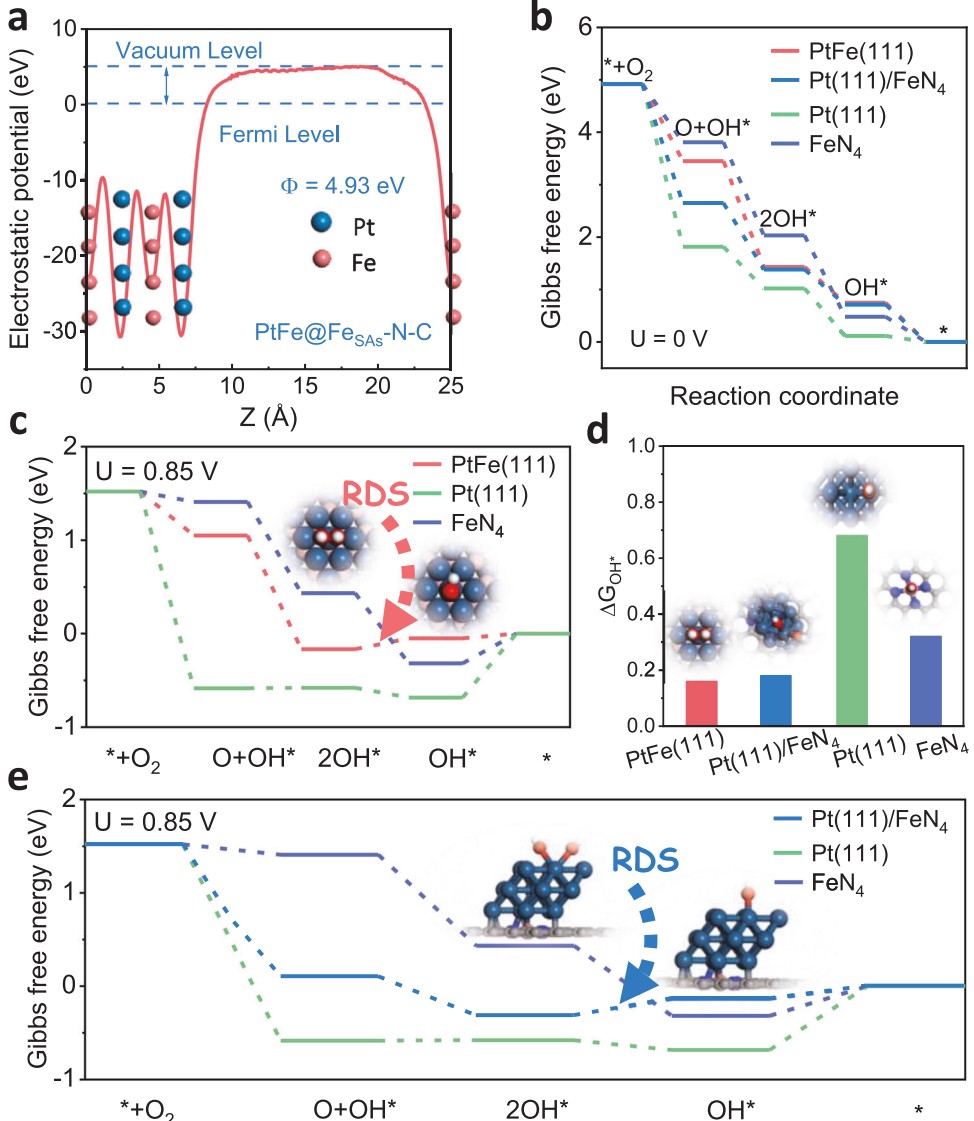

**Fig. 5 | Theoretical study the enhanced performance of PtFe@Fe_SAs-N-C electrocatalyst. a** The work function ($e_\Phi$) of PtFe (111). **b** Gibbs free energy diagram of ORR on PtFe (111), Pt (111)/ FeN$_4$, Pt (111), and FeN$_4$ at $U = 0$ V. **c** Gibbs free energy diagram of ORR on PtFe (111), Pt (111), and FeN$_4$ at $U = 0.85$ V. **d** Energy barrier for OH* protonation to form H$_2$O. **e** Gibbs free energy diagram of ORR on Pt (111)/ FeN$_4$, Pt (111), and FeN$_4$ at $U = 0.85$ V. (dark blue, rose red, blue, red, gray, and white spheres represent Pt, Fe, N, O, C, and H atoms, respectively).

shows no significant decay in current density after 30,000 continuous cycles, with low voltage decreases of only 6 mV (H$_2$-O$_2$) and 2 mV (H$_2$-air) at a current density of 0.8 A cm$^{-2}$; these results show the excellent durability of PtFe@Fe_SAs-N-C. The mass activity at 0.9 V$_{iR\text{-free}}$ slightly decreases to 0.73 A mg$_{Pt}^{-1}$ even after 30,000 cycles in H$_2$-O$_2$ environment; this corresponds to only a 3% decrease. Additionally, this result surpasses the 2025 DOE durability goal of less than 40% mass activity loss after 30,000 cycles[26]. A chronoamperometric test under H$_2$-O$_2$ at 0.6 V was also conducted for both the anode and cathode to further evaluate the long-term durability of the different cathode catalysts (Fig. 6f). The fuel cell with a PtFe@Fe_SAs-N-C cathode shows a nearly constant current density over 220 h in the H$_2$-O$_2$ environment. In contrast, the current density of the Pt/C fuel cell drops by 60% in 120 h. Both the square-wave potential-cycling and constant-voltage tests confirm that PtFe@Fe_SAs-N-C has better fuel cell durability than the Pt/C, Pt-based, and other hybrid electrocatalysts (Fig. 6g and Table S8)[6,7,23,26,46–51]. The TEM images after the H$_2$-O$_2$ stability test show (Fig. S50) that the original morphology and elemental distribution remain well preserved, and no obvious particle agglomeration occurs

after more than 220 h of operation, which further confirms the structural stability of PtFe@Fe_SAs-N-C catalyst.

## Discussion

In summary, we constructed a stable low-Pt-based electrocatalyst consisting of ultrafine PtFe nanocrystals on Fe_SAs-N-C with a well-optimized spin occupancy state for the ORR. With the proposed directional spin charge injection effect, the Pt $dz^2$ orbital occupation state of PtFe@Fe_SAs-N-C is well-regulated by the dual-Fe sites from the ordered PtFe alloys and an atomically dispersed Fe-N-C support. As a result, PtFe@Fe_SAs-N-C has a high mass activity (0.75 A mg$_{Pt}^{-1}$) at 0.9 V$_{iR\text{-free}}$, high peak power density of 1.24 W cm$^{-2}$, and an excellent rated power of 10.3 W mg$_{Pt}^{-1}$ during PEMFC tests. More importantly, PtFe@Fe_SAs-N-C demonstrates an extraordinary durability, with a high mass activity retention of 97%, a voltage loss of only 6 mV at 0.8 A cm$^{-2}$ after 30,000 cycles, and no noticeable current decrease at 0.6 V over 220 h. We developed a combination method using a real-time magnetic response recorder-electrochemically-reaction, *operando* spectroscopy, and theories to confirm the 4e$^-$ dissociative reaction pathway

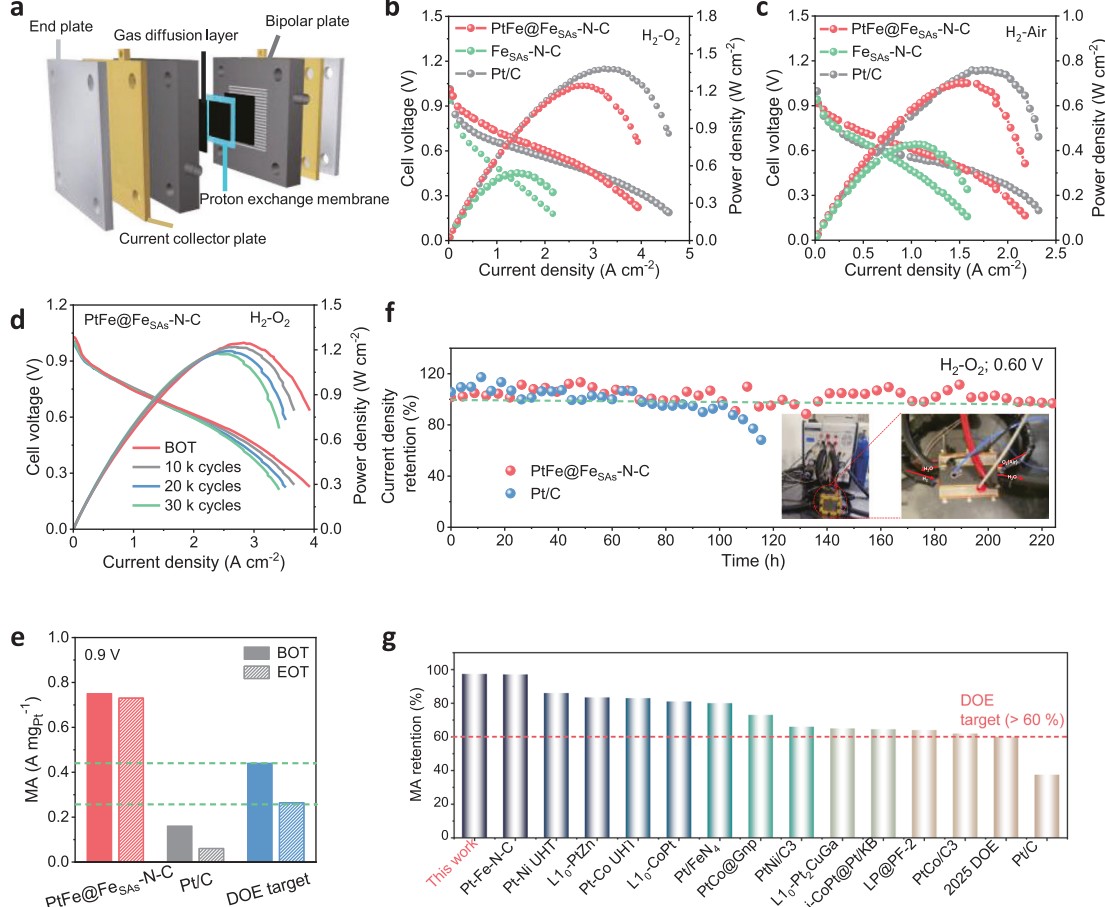

**Fig. 6 | Performance evaluation of the PtFe@Fe$_{SAs}$-N-C cathode in the fuel cell.**
**a** The structural components of a single PEMFC. **b** H$_2$/O$_2$ and **c** H$_2$-air fuel cell polarization (left axis) and power density (right axis) plots with loadings of 0.2 mg$_{Pt}$ cm$^{-2}$ for Pt/C, 0.12 mg$_{Pt}$ cm$^{-2}$ for PtFe@Fe$_{SAs}$-N-C, and 4 mg cm$^{-2}$ catalyst loading for Fe$_{SAs}$-N-C in the cathode under H$_2$/O$_2$ flow at 80 °C, 28 psi. **d** H$_2$/O$_2$ fuel cell polarization (left axis) and power density (right axis) plots of the PtFe@Fe$_{SAs}$-N-C

cathode before (BOT) and after 30,000 potential cycles between 0.6 and 0.95 V. **e** Mass activity (MA), MA test under H$_2$/O$_2$ flow at 80 °C, 28 psi. **f** Current density retention as a function of time for PtFe@Fe$_{SAs}$-N-C (0.12 mg$_{Pt}$ cm$^{-2}$) and Pt/C (0.2 mg$_{Pt}$ cm$^{-2}$) at a constant potential of 0.6 V under H$_2$/O$_2$ flow at 80 °C, 0 psi. **g** Comparison of MA retention after square wave accelerated durability testing between PtFe@Fe$_{SAs}$-N-C with the state-of-art in the literature.

through Griffiths adsorption; this method, addressed the limitations of difficulty of O-O bond activation and OH* site-blocking effects on the Pt sites, which accelerated the reaction kinetics and decreased the ORR overpotential to an ideal value. Moreover, bypassing H$_2$O$_2$ greatly improved the durability of PtFe@Fe$_{SAs}$-N-C. Our study provides many opportunities for designing low Pt catalysts, in which the selective evolution of different reactive oxygen species can be rationally tuned for highly efficient practical PEMFCs.

## Methods
### Chemicals and reagents
All chemicals were purchased and used without further purification. Iron(III) acetylacetonate (Fe(acac)$_3$, 98%, Innochem), zinc nitrate hexahydrate (Zn(NO$_3$)$_2$·6H$_2$O, 99%, Alfa Aesar), 2-methylimidazole (99%, Aladdin), methanol (AR), ethanol (AR) and isopropanol ((CH$_2$OH)$_2$, AR) were purchased from Sinopharm Chem. Reagent Co., Ltd. Nafion D-520 dispersion (5% w/w in water and 1-propanol, Alfa Aesar), commercial Pt/C (20 wt % Pt, Johnson Matthey), The ultrapure water (18.2 MΩ·cm) used in all experiments was obtained through ionexchange and filtration.

### Synthesis of Fe/ZIF-8 and ZIF-8
In the typical synthesis, 952 mg Zn(NO$_3$)$_2$·6H$_2$O and 88 mg Fe(acac)$_3$ were dissolved in 12 mL of methanol with stirring for 20 min, and then 24 mL methanol containing 1.05 g 2-Methylimidazole was added under

vigorous stirring for another 10 min at room temperature. Subsequently, the mixture was transferred into a 50 mL Teflon-lined stainless-steel autoclave for reaction at 120 °C for 4 h. After cooling to room temperature, the product was collected by centrifugation and washed with methanol four times, and finally dried at 60 °C in a vacuum oven. Individual ZIF-8 was synthesized by the same method without Fe(acac)$_3$.

### Synthesis of Fe$_{SAs}$-N-C and NC
The dried Fe/ZIF-8 powder was added into a ceramic boat and heated in a quartz tube at 950 °C for 2 h under an Ar atmosphere with a heating rate of 5 °C min$^{-1}$. The sample NC was prepared in the same procedure by adding ZIF-8.

### Synthesis of PtFe@Fe$_{SAs}$-N-C
The PtFe nano alloys were deposited on Fe$_{SAs}$-N-C[25,52]. At first, 16 mg Fe$_{SAs}$-N-C was dispersed into 8 mL of (CH$_2$OH)$_2$ solution containing 20 mg NaOH under continuous ultrasound. Then, 8 mL of (CH$_2$OH)$_2$ solution containing 10 mg H$_2$PtCl$_6$·6H$_2$O was added into the above uniformly dispersed black solution under continuous ultrasound, and then vigorously stirred with magnetic force for 30 min. The obtained mixed solution was heat treated in an oil bath at 115 °C for 2 h under continuous stirring. After natural cooling, the PtFe@Fe$_{SAs}$-N-C precursor were collected by filtration washed of the resulting solution with ultrapure water, and finally freeze-dried. In order to convert the

PtFe alloy precursor into structurally ordered PtFe intermetallic nanocrystals, the PtFe@Fe$_{SAs}$-N-C precursor was further annealed at 900 °C for 1 h under a flowing argon atmosphere at a heating rate of 10 °C min$^{-1}$, and then naturally cooled to room temperature to obtain PtFe@Fe$_{SAs}$-N-C.

## Physical characterizations

The structure of the samples was characterized by a transmission electron microscope (TEM, FEI Tecnai G220) and a field-emission scanning electron microscope (FE-SEM, JEORJSM-6700F). The HAADF-STEM images were obtained using a JEOL JEM-ARM200F at an accelerating voltage of 200 kV. Powder X-ray diffraction (PXRD) patterns were collected using a Y-2000X-ray diffractometer using copper Kα radiation ($\lambda = 1.5406$ Å) at 40 kV and 40 mA[52]. The X-ray photoelectron spectroscopy (XPS) measurements were performed with an ESCA LAB 250 spectrometer using a focused monochromatic Al Kα line (1486.6 eV) X-ray beam with a diameter of 200 μm. The Raman measurements were taken at 532 nm on a Renishaw Microscope System 16 RM2000. UPS measurements were also carried out on an ESCA LAB 250 Xi spectrometer with He I resonance lines (21.22 eV). The Fe K-edge and Pt L$_3$-edge XANES and the EXAFS were investigated at the BL14W beamline of Conducted at Shanghai Synchrotron Radiation Facility (SSRF) (Shanghai, China). In the fluorescence mode using a fixed-exit Si (111) double crystal monochromator. The Pt and Fe content was conducted on an inductively coupled plasma mass spectrometry (ICP-MS, Thermo Scientific iCAP™ RQ). The O$_2$-temperature-programmed desorption (TPD) of the samples was measured using Micromeritics AutoChem 2950 HP. The catalyst was first pretreated at 150 °C, purged with He for 2 h, cooled to room temperature. Then, purge with 5% O$_2$/He at 25 °C for 2 h. Finally, the desorption curve of O$_2$ was recorded online in He atmosphere.

## In-situ FTIR measurements

In-situ FTIR measurements refer to previous reports[53]. Detailed, In-situ surface-enhanced Fourier transform infrared absorption spectroscopy (FTIR) was employed on a Thermo Scientific Nicolet iS50 with the MCT detector. The original diagram of the device is shown in Fig. S51. In this test, the catalyst was loaded onto the surface of a silicon column with a gold-plated thin layer as the working electrode. In-situ electrochemical testing was conducted in 0.1 M HClO$_4$ saturated with O$_2$, using Pt net and Ag/AgCl electrode as the counter electrode and reference electrode, respectively. The test ranges from 0 V to 1.05 V, and the operation time for each potential is approximately 5 minutes. The spectra collected in OCP (without additional bias) are used as baselines. All spectra are analyzed after subtracting the baseline. According to the Nernst equation ($E_{RHE} = E_{Ag/AgCl} + 0.0592$ pH + 0.197 V), calibrate all potentials relative to RHE.

## *Operando* XAFS measurements

All XAFS data were collected at 1W1B station of Beijing Synchrotron Radiation Facility, China[15]. In this *operando* test, a customized three electrode system was used, and the original equipment diagram is shown in Fig. S52. The working electrode is composed of 1 × 1 cm$^2$ carbon paper loaded with catalyst, the counter electrode and reference electrode are carbon rod and Ag/AgCl electrode, respectively, and O$_2$ saturated 0.1 M HClO$_4$ solution is used as the electrolyte. The test potential ranges from 0 V to 1.1 V, covering the entire oxygen reduction reaction potential window. After running for 3 minutes at each potential, data spectral lines are collected. According to the Nernst equation ($E_{RHE} = E_{Ag/AgCl} + 0.0592$ pH + 0.197 V), calibrate all potentials relative to RHE.

## *Operando* magnetometry

The temperature magnetic susceptibility curves measurements, expose the sample to a temperature range of 5 to 300 K at a rate of 4 K/min, while applying an external field of 5 kOe[17]. The electrochemical measurements were carried out in an argon-filled glove box. Using a three-electrode system with a 1 × 0.3 cm$^2$ carbon paper loaded with PtFe@Fe$_{SAs}$-N-C catalyst as the working electrode, the counter electrode and reference electrode are Pt film and Ag/AgCl, respectively, and O$_2$-saturated 0.1 M HClO$_4$ solution as the electrolyte. According to the Nernst equation ($E_{RHE} = E_{Ag/AgCl} + 0.0592$ pH + 0.197 V), calibrate all potentials relative to RHE. The working electrodes obtained from each constant potential test are quickly transferred to the comprehensive physical property measurement system (PPMS, Quantum Design) under inert gas protection for magnetic performance testing. The magnetization values given in emu g$^{-1}$ are defined per unit weight of materials. The linear magnetic background signals from the carbon paper is deducted from the total magnetic moment.

## Electrocatalytic measurement

The electrochemical performance evaluation was conducted on the CH Instruments 760E electrochemical workstation[54]. All the electrochemical measurements were conducted in a three-electrode system, the counter electrode and reference electrode are Pt film and Ag/AgCl, respectively. According to the Nernst equation ($E_{RHE} = E_{Ag/AgCl} + 0.0592$ pH + 0.197 V), calibrate all potentials relative to RHE. The working electrode was obtained by mixing 2 mg of the catalyst in 1.0 mL of a solution containing 670 μL isopropanol, 290 μL ultra-pure water, and 40 μL 5% Nafion solution, followed by ultrasonic treatment for 30 min to form uniform inks that were drop-cast onto the GC-RDE and air-dried. The area of the GC working electrode is 0.19625 cm$^2$. 10 μL of catalyst ink was then pipetted on the GC surface, leading to a loading of 0.1 mg cm$^{-2}$. Pt/C (20 wt%) electrode was prepared by using the same procedure. 0.1 M HClO$_4$ solutions saturated with O$_2$ were employed as the electrolyte for ORR. The electrolyte is prepared and used immediately. Take a quantitative amount of concentrated perchloric acid (HClO$_4$, 70%, 99.999% trace metals basis, Sigma-Aldrich) and add it to a volumetric flask to dilute to 0.1 M before use.

All electrochemical measurements were conducted at room temperature (25 °C) using a rotating disk and rotating ring-disk electrode (RDE and RRDE, Pine Instrument)[55]. All data ensures repeatability and is presented in this work only after multiple measurements to ensure consistency. Linear sweep voltammetry (LSV) curves were obtained to investigate the performance of the catalysts at a scan rate of 5 mV s$^{-1}$ with a rotating speed of 1600 rpm, the LSV curves were recorded without *iR* compensation, and all currents were corrected by deducting the background current that measured in N$_2$-saturated electrolyte. Electrochemical impedance spectroscopy (EIS) is measured at open circuit voltage with a low frequency of 0.1 Hz, a high frequency of 1000000 Hz, and an amplitude of 0.005 V. The resistance values of the catalysts are the average of the electron transfer resistance obtained through multiple EIS tests.

The accelerated durability test was conducted in O$_2$-saturated 0.1 M HClO$_4$ solution at a scan rate of 50 mV s$^{-1}$ with a potential between 0.6 V and 1.0 V vs. RHE for 30,000 and 50,000 cycles.

The electron transfer number ($n$) and kinetic current density ($J_K$) were calculated from the Koutecky-Levich (K-L) equation[56]:

$$\frac{1}{J} = \frac{1}{J_L} + \frac{1}{J_K} = \frac{1}{B\omega^{0.5}} + \frac{1}{J_K} \tag{1}$$

$$B = 0.62nFC_0(D_0)^{2/3}\nu^{-1/6} \tag{2}$$

$$J_K = nFkC_0 \tag{3}$$

Where $J$ is the measured current density. B is determined from the slope of the K-L plot. $J_L$ and $J_K$ are the diffusion-limiting current densities and kinetic current densities, n is the transferred electron

number, $F$ is the Faraday constant ($F = 96485\,C\,mol^{-1}$), $C_O$ is the $O_2$ concentration in the electrolyte ($C_O = 1.26 \times 10^{-6}\,mol\,cm^{-3}$), $D_O$ is the diffusion coefficient of $O_2$ ($D_O = 1.93 \times 10^{-5}\,cm^2\,s^{-1}$), and $\upsilon$ is the kinetic viscosity ($\upsilon = 0.01009\,cm^2\,s^{-1}$).

The $HO_2^-$% and transfer number ($n$) were calculated by the following equations[57]:

$$HO_2^-\% = 200 \times \frac{I_R/N}{I_D + I_R/N} \tag{4}$$

$$n = 4 \times \frac{I_D}{I_D + I_R/N} \tag{5}$$

where $I_D$ and $I_R$ are disk current and ring current, respectively. $N = 0.4$ represents the ring collection efficiency.

### Calculation of unpaired d electrons for Fe ions in catalysts

According to the $\mu_{eff} = \sqrt{n(n+2)}\mu_B$ equation, evaluate the unpaired electron number (n) of Fe in the PtFe@Fe$_{SAs}$-N-C and Fe$_{SAs}$-N-C from the effective magnetic moment ($\mu_{eff}$). The $\mu_{eff}$ was evaluated based on the Langevin theory, $\mu_{eff} = \sqrt{8C}\mu_B$, where $C$ is the Curie constant and the slope of the curve of $\chi^{-1}$-$T$ according to the Curie-Weiss law $\chi = C/(T-\Theta)$, $\chi$ and $\Theta$ are the susceptibility and Curie-Weiss temperature, respectively. The $\chi$ can be obtained derived from the $\chi = M/H$ equation, where $M$ and $H$ are the magnetization and magnetic field intensity, respectively.

### Fenton-like reactivity measurements

The production of intermediate hydrogen peroxide was evaluated via a reported method[37]. Briefly, the metal ions were ultrasonically dissolved in 0.1 M $HClO_4$ solution respectively to achieve a 40 μg mL$^{-1}$ homogeneous suspension. The desired amounts of 2,20-azinobis(3-ethylbenzthiazoline-6-sulfonate) (ABTS), and hydrogen peroxide were sequentially added to reach the concentrations of 2 mM and 20 mM, respectively. After 5 min, the solution was diluted with 0.1 M $HClO_4$ (3:100) and characterized by UV-Vis spectroscopy (SHIMADZU UV-1900 Series UV-Vis Spectrophotometer).

### Fabrication and tests of $H_2$-$O_2$ fuel cells

The cathode catalyst ink includes a catalyst, Nafion (5%), isopropanol, and ultrapure water[58]. After mixing, it is sonicated in an ice water bath for 2 h to form a uniform mixture, which is then sprayed onto a gas diffusion layer (GDL).The Pt@Fe$_{SAs}$-N-C cathode with 0.12 mg cm$^{-2}$ Pt loading. The Fe$_{SAs}$-N-C cathode with a 4 mg cm$^{-2}$ catalyst loading and a commercial Pt/C cathode with a 0.2 mg cm$^{-2}$ Pt loading were prepared using the same protocol. Commercial Pt/C-coated GDL with a 0.2 mg cm$^{-2}$ Pt loading was used as the anode. Use the commercial fuel cell testing system (SMART2) to evaluate fuel cell performance in a single cell. The MEA was sandwiched between two graphite plates with single serpentine flow channels. The cell was operated at 80 °C with a back pressure of 28 psi. Pure $H_2$ and $O_2$/air, with 100% relative humidity (RH), were supplied to the anode and cathode at a gas flow rate of 300-800 ml min$^{-1}$. The polarization curves were recorded at $H_2$ and $O_2$/air flow rates of 300 and 400 ml min$^{-1}$, respectively, and 28 psi back pressures. Square-wave potential cycles are performed at 0.6 to 0.95 V with a sweep speed of 50 mV s$^{-1}$, following the DOE protocol for evaluating the durability of catalysts[7]. Durability was assessed by chronoamperometric measurements at a constant discharge voltage of 0.6 V, $H_2$/$O_2$ flow rate of 300/400 ml min-1, 0 psi back pressure. Rated power measured at 0.67 V[49,59].

### DFT calculations

The spin-polarized density functional theory (DFT) calculations were conducted using the Vienna ab initio simulation package (VASP)[60] with the projector augmented wave (PAW)[61] pseudopotentials. The Hubbard-U correction (DFT + U method) was applied to improve the description of localized metal $d$-electrons in the single atom Fe catalysts[62]. Van der Waals interaction was described by using the DFT-D3 method[63]. The electron exchange correlation function was described within the generalized gradient approximation (GGA)[61] with the Perdew-Burke-Ernzerhof (PBE) functional. The cut-off energy of 450 eV for plane-wave was set for all calculations while the atomic position was fully relaxed until the residual forces was less than 0.02 eV/Å. The Brillouin zone was sampled with a 3 × 3 × 1 k-points based on the Gamma-centered Monkhorst-Pack mesh. A vacuum space exceeding 30 Å was employed to avoid the interaction between two periodic units in z direction. The Fe/Pt 2 × 2 × 1 supercell was built based on Pt (111) orientation crystal where Pt atoms in one layer were replaced by Fe atoms.

The computational hydrogen electrode (CHE) model was used to calculate the free energy of reactions involving electron-proton transfer. According to the method presented by Nørskov[64], the Gibbs free energy diagrams for ORR were calculated by the equation:

$$\Delta G = \Delta E + \Delta ZPE - T\Delta S \tag{6}$$

where $\Delta E$ is the thermodynamic energy difference of reactants and products; $\Delta ZPE$ and $\Delta S$ are the energy difference of zero-point energy and entropy; $T$ is the temperature and 298.15 K is employed.

The free energy of $H^+$ ions has been corrected by the concentration dependence of the entropy:

$$G(pH) = -kT\ln[H^+] = kT\ln10*pH \tag{7}$$

(0.059526 for 0.1 M $HClO_4$; 0.773844 for 0.1 M KOH).

## Data availability

All data needed to evaluate the conclusions in the paper are present in the paper and/or the Supplementary Information. Source data are provided as a Source Data file. Source data are provided with this paper.

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

## Acknowledgements

This work was financially supported by the Key Projects of the National Natural Science Foundation of China (U22A20107), the Key Projects of the Henan Provincial Science and Technology R&D Program Joint Fund (222301420001), the Distinguished Young Scholars Innovation Team of Zhengzhou University (32320275). Acknowledge the Beijing Synchrotron Radiation Facility (BSRF) 1W1B station for XAS measurements. Li-Rong Zheng is grateful to the support of XAS.

## Author contributions

D.P.X., M.H.S., and J.N.Z. conceived the project. D.P.X. carried out the synthesis, most of the structural characterizations, electrochemical tests, and fuel cell measurements. Y.Y. and Y.F.W. performed the DFT calculations. Y.F.Y. and H.Z.G. performed the *operando* magnetometry characterizations. S.R.X., J.Q.Z., M.H.S., and J.N.Z. discussed the results and commented on the manuscript. D.P.X. and J.N.Z. analyzed the data and co-wrote the paper. J.N.Z. supervised the project.

## Competing interests

The authors declare no competing interests.
