## [Peer Review File · Nature Communications]

Spin occupancy regulation of Pt d-orbital for a robust low-Pt catalyst towards oxygen reductionREVIEWER COMMENTS

Reviewer #1 (Remarks to the Author):

In this work, Zhang and coworkers reported the synthesis of PtFe nanoparticles on Fe single atom-doped porous carbons for the oxygen reduction reaction. These materials are characterized by TEM, EELS images, XAFS, etc. It also shows the good performance for the ORR in 0.1 M HClO₄. A practical PEMFC was also constructed using PtFe nanoparticles. However, I feel that this present research cannot meet the standard of Nature Communications based on the following comments. (1) The authors claimed that the orbital occupation state of PtFe was regulated by dual-Fe sites from PtFe alloys and atomically dispersed FeNC support. However, the constructed composites composed of PtFe nanoparticles, Fe single atoms, and carbons support are too complex. So it is not reasonable that the enhanced ORR is ascribed to spin state regulation. I believe that lots of carbon defects exist in this composite. How did the authors exclude the effects of carbon defects and vacancies on the ORR activity enhancements.

(2) The PtFe nanoparticles on FeNC for ORR have been reported in ACS Nano 2024, 18, 1, 551–559. This research is quite similar as this reference. In this reference, FeNC active sites electronically coupled with PtFe alloys (PtFe-FeNC) were successfully prepared by a vapor deposition strategy as an ultralow Pt loading (0.64 wt %) hybrid electrocatalyst. The FeN₄ sites on the FeNC matrix are able to effectively anchor the PtFe alloys, thus inhibiting their aggregation during long-life cycling. These PtFe alloys, in turn, can efficiently restrain the leaching of the FeN₄ sites from the FeNC matrix. Based on the results in the reference, it was found that the lack of the novelty in present research prevents its publication in Nature Communications.

(3) The FeNC support played a key role of adjusting PtFe nanoparticles for the ORR. The more details of FeNC in this composite should be made. The contents of the FeNC should be confirmed and adjusted to study its effect on the PtFe. In fact, it is difficult to make clear the state of Fe single atoms in porous carbons. So, precise regulation using this FeNC is not believable. I suggest that the authors could use of iron phthalocyanine with well-defined structures for studying spin state effects.

Reviewer #2 (Remarks to the Author):

In this work, Zhang et al. constructed an ultra-low-Pt-based electrocatalyst consisting of ultrafine PtFe NCs on FeSAs-N-C matrix with well-optimized spin occupancy state for ORR. Importantly, the 4e⁻ dissociation pathway was confirmed through a series of in-situ characterization, especially the interesting operando magnetometry to confirm the important role of the ferromagnetic sites of PtFe/Fe-N-C in solving the challenges of O-O bond activation and OH* site-blocking effect on Pt sites, achieving an ideal ORR overpotential. It can be published in Nature Communication after some revisions as follows:

1. What is the role of FeSAs-N-C support in PtFe@FeSAs-N-C catalyst in promoting the stability and activity for ORR. Please provide the corresponding evidence and analysis.
2. In this paper, the authors propose that Fe injection charge into the d orbitals of Pt can optimize the spin configuration and adsorption behavior of Pt sites. However, in PtFe@FeSAs-N-C catalysts, Fe exists in both support and alloy nanoparticles. Does the Fe injection of charge into the d orbitals of Pt come from the carrier or the alloy nanoparticles?
3. It is known that the biggest problem of Fe-N-C catalyst in proton exchange membrane fuel cells is Fe shedding. Whether it is taken into account in this work, please conduct necessary analysis and proof.
4. It is good for the authors to propose the O₂-side adsorption on the Pt sites of dissociation pathway for the PtFe@FeSAs-N-C. The author should provide more sufficient evidence to directly demonstrate such mechanism, such as the corresponding calculations.
5. Page 8, "As Fe atoms are alloyed with Pt, and N-Fe bonds are broken, resulting in a significant content decrease of N-Fe and pyridinic N." This analysis is inconsistent with Table S4, please confirm.
6. In Figure 6, TEM images of PtFe@FeSAs-N-C in the fuel cells after stability test results should also be provided and discussed.

Reviewer #3 (Remarks to the Author):

Xue et al. developed PtFe alloy/Fe-N-C material (PtFe@FeSAs-N-C) as an ORR catalyst for proton exchange membrane fuel cell (PEMFC) application. Extensive ex-situ and in-situ characterizations were conducted to investigate the material's structure and electronic properties. PtFe@FeSAs-N-C exhibited high activity, surpassing commercial Pt/C, and robust stability under accelerated durability testing and chronoamperometric characterization. The authors proposed an intriguing concept of the directional spin-charge injection effect to elucidate the enhanced catalytic activity. However, the justification of the authors' claims appears limited, and the data interpretation is somewhat biased.

1. Firstly, a concern arises regarding the inconsistency in the direction of electron redistribution between PtFe nanoparticles (NPs) and the FeN₄ support. The shift of the Pt 4f XPS peak to a higher binding energy and the increase of Pt L₃-edge absorption intensity suggest "electron transfer from Pt to the FeSAs-N-C support". Conversely, Kelvin probe force microscopy (KPFM) and DFT calculations suggest "electrons tend to flow from the FeSAs-N-C support to PtFe NCs". The authors adopted the latter interpretation (i.e., Pt as the electron acceptor) to explain the enhanced catalytic performance. This inconsistency may stem from the dependency of Pt NPs electronic properties on NP size, as demonstrated by Yan et al. (Nat Commun 10, 4977 (2019)) and Wei et al. (Angew. Chem. Int. Ed. 2021, 60, 16622). KPFM and DFT calculations represent local characterization, focusing on specific NPs with particular sizes, whereas XPS (limit to 10 nm thickness) and XAS provide more "bulk" characterizations, probing multiple Pt NPs with varying sizes. Hence, it can be inferred that in PtFe@FeSAs-N-C, the dominant Pt species is more likely to be electron-donating than electron-accepting, contradicting the authors' claims. The authors should provide a more comprehensive discussion addressing this inconsistency.

2. Secondly, the impact of 'spin charge injection' on catalytic activity remains uncertain. It is unclear whether the spin configuration of the active site is the determinant of activity or simply a dependent variable that fluctuates throughout the reaction. Additionally, it seems that orbital occupancy plays a more crucial role in determining catalytic activity compared to spin characteristics.

- The authors assert that the spin charge injection effect optimizes the activation and dissociation of paramagnetic O₂ molecules (Page 18, line 381). However, both experimental and computed results suggest otherwise. O₂ temperature-programmed desorption (Figure S35) indicates weaker O₂ adsorption on PtFe@FeSAs-N-C compared to FeSAs-N-C. This weaker O₂ adsorption, resulting in less back donation of electrons from Pt 5d to O₂ π*, leads to slower activation/dissociation, which is unfavorable for the diffusion-limited ORR. Furthermore, energy pathway analyses (Figures 5b, c, e, and S40) demonstrate that the dissociation of O₂ and the formation of O + OH* are much more thermodynamically favorable, with a significant decrease in free energy, on Pt(111) surface than on the spin-injected PtFe(111) surface.

- The enhanced activity is also attributed to the facilitated protonation of OH*. Increased orbital occupancy weakens the Pt-OH* bonding, thereby promoting OH* protonation (rate-determined step). However, the effect of spin injection remains unclear. PtFe(111) and Pt(111)/FeN₄, which likely possess different spin and magnetic moments, exhibit very similar reaction energy barriers of 0.16 and 0.18 eV, respectively.

- Page 14, line 293, the authors attribute the decrease in magnetization and effective magnetic moment of PtFe@FeSAs-N-C as the applied potential reduces from 1.1 to 0.2 VRHE, the Griffiths adsorption O₂ molecules and the optimization of O₂ molecules/intermediates adsorption behavior on the active Pt site. However, this assertion warrants further evaluation. Why does the adsorption of O₂ in an on-top configuration on a single Pt atom reduce the magnetic moment? Furthermore, it's important to note that Griffiths adsorption often leads to an associative ORR pathway rather than the dissociative pathway as claimed (Angew. Chem. 2023, 135, e202301833).

3. The authors assert that the injection of electrons from Fe to Pt "... also restricts catalyst coalescence and retards the oxidative dissolution, diffusion, and Ostwald ripening processes" (Page 5, Lines 82-84). However, this claim lacks robust support.

- Figure S30a (HAADF-STEM of the post-ADT sample) appears to depict the aggregation of two PtFe NPs, with a size considerably larger than what is claimed ("3.35 nm without apparent agglomeration" Page 13 line 268). It is recommended to provide images with a larger view, capturing more nanoparticles to substantiate the absence of coalescence/agglomeration and Ostwald ripening.

- It is quite surprising that the PtFe nanoparticles (NPs) can withstand the oxidative O₂-saturated acidic conditions without experiencing any dealloying or dissolution, which are typically observed in Pt-alloy ORR electrocatalysts. This is particularly intriguing considering the high oxophilicity and low redox potential of Fe. To bolster their claims, the authors should conduct further characterization studies, ICP-MS of the remaining electrolytes, ICP/EDS, XRD, and XPS of the used electrocatalyst, to elucidate any potential dissolution and oxidation phenomena.

- Optionally, the authors may consider calculating the coherence/bonding energy of PtFe to elucidate its structural stability. Additionally, conducting electron localization function (ELF) analyses could help illustrate the "strong metal-support interaction" between PtFe nanoparticles (NPs) and the FeN₄ support, offering a mechanistic understanding of the material's robust stability.

- It is well-known that the start-up/shutdown procedure in fuel cells can induce potential excursions, leading to catalytic deterioration that is more severe than under normal load condition (0.6-1 V_{RHE}). Optionally, the authors may further assess the stability of materials under start-up/shutdown conditions (1-1.5 V_{RHE}) following the DOE-suggested protocol.

4. The density functional theory (DFT) analysis appears to lack comprehensiveness. It is unclear why the authors did not explore the model of PtFe(111)/FeN₄, which would reflect the structure of PtFe/FeSAs-N-C.

5. Page 11, line 211, the statement "the decrease of the magnetic moment of Fe atoms with low Pt loading can be attributed to the spin charge injection effect between Fe and Pt sites..." lacks clarity regarding how spin redistribution could reduce the average magnetic moment. Alternatively, the decrease in magnetic moment may stem from the decrease in Fe content in PtFe@FeSAs-N-C compared to FeSAs-N-C (Table S2).

6. Importantly, the authors should not assume that the electron donation from Fe sites in PtFe NCs and FeSAs-N-C impose a t_{2g}⁶e_g³ electron configuration on Pt atoms. The orbital occupation of t_{2g} and e_g orbitals in Pt should be determined through fitting analysis of Pt L₃-edge XANES.

7. Figure 1b inset, the particle size distribution of PtFe NPs was analyzed in a very narrow range (2-3 nm), leading to the claim of "excellent size uniformity of ~2.46 nm". The TEM and HAADF-STEM images (Figure 1c, S5, and S6), however, reveal the existence of numerous nanoparticles with much larger sizes (5-10 nm) and lower uniformity.

8. Using simulated STEM images (Figure 1e, Figure S31b) as evidence to study lattice structure should be approached with caution. The STEM reconstruction algorithm may introduce significant artifacts, especially when dealing with blurry HAADF-STEM images (Figure 1c, S31a). To ensure accuracy, the authors may consider providing clearer HAADF-STEM or ACFM images.

9. Figure S15 and Table S5 (57Fe Mössbauer spectroscopy), the percentage of iron metal (α and γ -Fe) appears to be relatively high, exceeding 30%. The authors should explain the formation of these components and elucidate whether they have any impact on the catalytic performance.

10. In PEMFC characterization, it is unclear why the authors did not use the same Pt loading for Pt/C and PtFe/FeSAs-N-C.

11. Minor comments:

- Page 3, lines 56-57, the mention of "...introducing new variables (second adsorption site)." may not be entirely relevant, as the O₂ adsorption is the on-top adsorption (Griffiths model).

- Figure S2 (PXRD) unveiled a shift in the diffraction peaks of PtFe@FeSAs-N-C towards lower 2 θ

(e.g., (001) and (111)), suggesting the presence of tensile strain. The authors should comment on the impact of this strain on the material's physicochemical properties and catalytic performance.

- Page 4, line 61, "proton coupling and electron transfer steps" should be "proton-coupled electron transfer steps".

- Figure 1b caption, "the coexistence with FeN₄ sites" should be removed as the atomic FeN₄ cannot be observed in TEM image.

- The caption of Figure 1d inset should be provided.

- Figure 1f (FFT pattern), the authors may consider providing a clearer image.

- Figure 1h, "d(002) should be "d(111)" as planes (002) and (001) are parallel.

- Table S4, the row label seems to be incorrect, as the contents of N-Fe and pyridinic N in PtFe@FeSAs-N-C are higher than those in FeSAs-N-C.

- Figure 3d, "inset: the absorbance value of the solution at 417 nm" is missing.

- Page 10 line 205, is it worth mentioning the presence of "one eg electron (t_{2g}4eg₁), the medium-spin Fe(III)N₄" considering that FeSAs-N-C contains significantly more medium-spin Fe(III)N₄ than PtFe@FeSAs-N-C.

- Figure 6b caption, "4 mgPt cm⁻² catalyst loading for FeSAs N-C" should be "4 mg cm⁻² catalyst loading for FeSAs N-C".

- Page 19 line 390, the meaning of "combining with the low reaction energy barrier of Pt(111)/FeN₄ can effectively improve the ORR kinetics of PtFe@FeSAs-N-C and make it have superior stability" is unclear. Why the low energy barrier in the Pt(111)/FeN₄ would benefit the kinetics of PtFe@ FeSAs-N-C?

- Section Methods/Electrocatalytic measurement, the mentions of "alkaline medium", "0.1 M KOH" and the potential range "between 0.15 V and 1.05 V vs. RHE for 10000 cycles" in stability test may be typos.

- The authors should also revise the manuscript to correct typos and grammatical errors. Importantly, any claims that are not been derived from authors' results should be cited. For instance. the source of 2025 DOE target should be cited.

Responses to reviewers' comments:

To reviewer #1:

In this work, Zhang and coworkers reported the synthesis of PtFe nanoparticles on Fe single atom-doped porous carbons for the oxygen reduction reaction. These materials are characterized by TEM, EELS images, XAFS, etc. It also shows the good performance for the ORR in 0.1 M HClO₄. A practical PEMFC was also constructed using PtFe nanoparticles. However, I feel that this present research cannot meet the standard of Nature Communications based on the following comments.

Reply: Thanks for the reviewer for sparing your precious time to review the manuscript. The reviewer's comments were very helpful overall, and we are appreciative of such constructive feedback on our manuscript. We have now revised the manuscript carefully based on your concerns. The main modifications and additions are as follows:

1. In order to provide a more comprehensive explanation of the source of activity and the influence of defect structures, we synthesized Pt@N-C catalysts and compared PtFe@FeSAsN-C, Pt@N-C, FeSAsN-C, and N-C catalysts through polarization curve testing, SCN⁻ poisoning experiments, and Raman spectroscopy characterization. It was confirmed that the enhancement of ORR activity is mainly attributed to the electronic structure interaction between Pt and Fe.
2. In order to understand more details of Fe-N-C in composite materials. On the one hand, by using the same synthesis method, the Fe content in Fe-N-C was regulated to study its impact on PtFe. On the other hand, composite catalysts were synthesized using FePc with different contents as Fe sources to explore ORR activity.
3. In order to comprehensively confirm the stability of PtFe catalyst, firstly, TEM characterization was carried out on the PtFe@FeSAsN-C sample after fuel cell durability testing, which still maintained the original morphology structure and element distribution well, and no obvious agglomeration phenomenon occurred. Secondly, we analyzed the metal content and carbon defect structure of the samples before and after durability testing through XPS, Raman spectroscopy, and ICP testing, fully confirming the stability of the PtFe@FeSAsN-C catalyst structure and its ability to effectively suppress metal detachment. Finally, we evaluated the stability of PtFe@FeSAsN-C under high potential (1.0-1.5 V) in

H₂-O₂ fuel cells based on the DOE control protocol, which still showed the highest mass activity retention rate.

4. we constructed a PtFe(111)/FeN₄ structural model to theoretically verify the experimental results. The calculation results confirm that, as confirmed in the experiment, among all structural models, the lowest rate-determining step energy barrier of PtFe (111)/FeN₄ structure is only 0.033 eV at U=0.85 V, indicating that this structure is more favorable for 4e⁻ ORR.

A list of point-to-point response was prepared, as shown in below.

Q1. The authors claimed that the orbital occupation state of PtFe was regulated by dual-Fe sites from PtFe alloys and atomically dispersed FeNC support. However, the constructed composites composed of PtFe nanoparticles, Fe single atoms, and carbons support are too complex. So it is not reasonable that the enhanced ORR is ascribed to spin state regulation. I believe that lots of carbon defects exist in this composite. How did the authors exclude the effects of carbon defects and vacancies on the ORR activity enhancements.

Reply: Thank you very much for your valuable review. Based on your consideration, in order to more comprehensively explain the source of activity and the influence of defect structure, we have synthesized Pt@N-C catalyst by preparing PtFe@Fe_{SAs}-N-C. Polarization curve test showed that the activity of PtFe@Fe_{SAs}-N-C catalyst ($E_{1/2}=0.872$) was significantly higher than that of Pt@N-C ($E_{1/2}=0.705$), Fe_{SAs}-N-C ($E_{1/2}=0.770$) and N-C ($E_{1/2}=0.438$) (**Figure R1a**). At the same time, the activity of PtFe@Fe_{SAs}-N-C catalyst before and after poisoning was tested using SCN⁻ poisoning single atom Fe sites. As shown in **Figure R1b**, the poisoned $E_{1/2}$ decreased by 56 mV, but still higher than Fe_{SAs}-N-C, indicating that the increase in activity was mainly due to the introduction of Pt on Fe_{SAs}-N-C support.

In order to further eliminate the influence of carbon defect structure differences among the four catalysts on catalytic activity, we tested the Raman spectra of the four samples, as shown in **Figure R1c**. After loading metals on N-C, the I_D/I_G values of PtFe@Fe_{SAs}-N-C, Pt@N-C, and Fe_{SAs}-N-C were comparable, proving that they have similar carbon defect structures.

In summary, the enhancement of ORR activity should be attributed more to the electronic structure interaction between Pt and Fe.

Figure R1. (a) ORR polarization curves of PtFe@FeSAs-N-C, Pt@N-C, FeSAs-N-C, N-C and Pt/C in O₂-saturated 0.1M HClO₄ at a rotation rate of 1600 rpm (scan rate = 5 mV s⁻¹). (b) LSV curves of PtFe@FeSAs-N-C in 0.1 M HClO₄ before and after the addition of SCN⁻. (c) Raman spectra of PtFe@FeSAs-N-C, Pt@N-C, FeSAs-N-C and N-C.

Q2. The PtFe nanoparticles on FeNC for ORR have been reported in ACS Nano 2024, 18, 1, 551-559. This research is quite similar as this reference. In this reference, FeNC active sites electronically coupled with PtFe alloys (PtFe-FeNC) were successfully prepared by a vapor deposition strategy as an ultralow Pt loading (0.64 wt %) hybrid electrocatalyst. The FeN₄ sites on the FeNC matrix are able to effectively anchor the PtFe alloys, thus inhibiting their aggregation during long-life cycling. These PtFe alloys, in turn, can efficiently restrain the leaching of the FeN₄ sites from the FeNC matrix. Based on the results in the reference, it was found that the lack of the novelty in present research prevents its publication in Nature Communications.

Reply: Thank you very much for your valuable review. According to your suggestions, we compared the novelty and differences between our research work and Sun's work with the following aspects.

In this reference, Sun et al. prepared FeN₄ active sites electronically coupled with PtFe alloys (PtFe-FeNC) catalysts by a vapor deposition strategy. There is strong electron coupling between FeN₄ active site and PtFe alloy. This strong electron coupling allows the FeN₄ site on the FeNC matrix to effectively anchor the PtFe alloy, thereby inhibiting their aggregation during long-term cycling. These PtFe alloys, in turn, can effectively restrain the leaching of FeN₄ sites from the FeNC matrix. Finally, the dual improvement of fuel cell activity and durability with ultralow Pt load was achieved.

In our work, Fe-N-C supported PtFe alloy nanoclusters (PtFe@FeSAs-N-C) catalysts were prepared by double-confined in-situ alloying method. We not only proved the existence of strong electron interaction between Fe-N-C support and PtFe alloy by XPS and *ex-situ* Fe K-

edge, Pt L₃-edge XAFS. More importantly, 1) we reveal a direct 4e⁻ electron reaction pathway at the PtFe@Fe_{SAs}-N-C surface site by *In-situ* ATR-FTIR, which can bypass the generation of H₂O₂ byproducts; 2) real-time magnetic response records-electrochemical reaction technology combined with *operando* XAFS characterization revealed that during the ORR process, a charge injection effect occurs between Fe and Pt in the catalyst, increasing the occupied state of Pt 5d orbital electrons, optimizing the adsorption of oxygen-containing species, weakening the O-O bond. In particular, there is an additional σ^* bond between the π orbital of the OH* intermediate, which leads to the weakening of the Pt-OH* bond strength, inhibits the steric effect, and effectively promotes the ORR kinetics. To our knowledge, such a characterization technique combined with analysis is also reported for the first time in electrocatalytic reactions, providing strong evidence for revealing orbital electron interactions between the active site and reaction intermediates.

Summary, the two works provide theoretical guidance for the rational design of low Pt catalysts with high activity and stability. Our work further revealed the ORR reaction pathway on the catalyst surface through a series of *in-situ* characterization, as well as the electron spin orbit interaction between the catalytic active center Pt and the oxygen reduction intermediates during the ORR process, providing a deeper understanding of the ORR process and reaction mechanism.

Q3. The FeNC support played a key role of adjusting PtFe nanoparticles for the ORR. The more details of FeNC in this composite should be made. The contents of the FeNC should be confirmed and adjusted to study its effect on the PtFe. In fact, it is difficult to make clear the state of Fe single atoms in porous carbons. So, precise regulation using this FeNC is not believable. I suggest that the authors could use of iron phthalocyanine with well-defined structures for studying spin state effects.

Reply: Thank you very much for your thoughtful suggestion. We fully agree with your point of view that FeNC support played a key role of adjusting PtFe nanoparticles for the ORR. Most studies on similar hybrid catalysts generally attributed the improvement of ORR performance to the strong metal-support interaction, but more specific and in-depth understanding of the evolution of ORR and the orbital electron interaction between the reaction site and the oxygen-containing intermediate state in the reaction process are still very limited. ORR involves a

multi-step proton coupling electron transfer process, involving spin evolution from paramagnetic triplet oxygen molecules to diamagnetic singlet oxygen intermediates. Therefore, the purpose of our work is to reveal the catalytic evolution mechanism at the electronic level while achieving high activity and stability, providing guidance for the preparation of efficient catalysts and the large-scale commercialization of fuel cells.

According to your suggestion, we have adjusted the Fe content in Fe-N-C to be 0 times, 0.5 times and 1.5 times of the current Fe_{SAs}-N-C. Three catalysts, Pt@N-C, PtFe@Fe_{SAs}-N-C-0.5, and PtFe@Fe-N-C-1.5, were prepared by the method of PtFe@Fe_{SAs}-N-C synthesis. Phase analysis and ORR activity evaluation were conducted on the prepared catalyst, and the results are shown in **Figure R2**. A decrease or increase in Fe content can lead to a decrease in the performance of the hybrid catalyst. The former may be due to the reduction of ordered alloy nanoclusters and weakened metal support interactions, while the latter may be due to the formation of inactive Fe compounds due to excessive Fe.

Meanwhile, thank you very much for your suggestion (use iron phthalocyanine with well-defined structures for studying spin state effects). However, high temperature heat treatment is necessary to form efficient ordered PtFe alloy nanoclusters. Even if FePc with a clear structure is used as Fe source, the coordination structure will still be destroyed during high temperature pyrolysis. This is the inevitable problem in the synthesis of such efficient catalyst synthesis methods, and it is also a problem that we have been and will solve in the future. Nevertheless, we prepared a series of corresponding PtFe@Fe-N-C catalysts synthesized with different amounts of FePc as carbon sources, and the activity evaluation results are shown in **Figure R3**. The catalytic activity of the catalysts synthesized with FePc as Fe source is much lower than that of commercial Pt/C, so it is even less convincing to study the catalytic mechanism with them.

Figure R2. (a) LSV curves of the prepared sample in O_2 -saturated 0.1M $HClO_4$ at a rotation rate of 1600 rpm. (b) PXRD pattern for the prepared sample. (c) Corresponding to the half wave potential of all catalysts in Figure R2a.

Figure R3. (a) LSV curves of Fe-N-C loaded PtFe alloy nanoparticles catalysts prepared with different contents of Pt:Fe as the Fe source. (b) Corresponding to the half wave potential of all catalysts in Figure R3a.

To reviewer #2:

In this work, Zhang et al. constructed an ultra-low-Pt-based electrocatalyst consisting of ultrafine PtFe NCs on Fe_{SAs}-N-C matrix with well-optimized spin occupancy state for ORR. Importantly, the 4e⁻ dissociation pathway was confirmed through a series of in-situ characterization, especially the interesting operando magnetometry to confirm the important role of the ferromagnetic sites of PtFe/Fe-N-C in solving the challenges of O-O bond activation and OH* site-blocking effect on Pt sites, achieving an ideal ORR overpotential. It can be published in Nature Communication after some revisions as follows:

Reply: Thank you for your recognition and valuable comments on our manuscript. We have now amended the manuscript carefully based on your concerns. A list of point-to-point response is as below.

Q1. What is the role of Fe_{SAs}-N-C support in PtFe@Fe_{SAs}-N-C catalyst in promoting the stability and activity for ORR. Please provide the corresponding evidence and analysis.

Reply: Thank you very much for your valuable review. Based on your suggestion, we have combined experimental evidence to summarize the role of Fe_{SAs}-N-C support in detail as follows:

- 1) **As an Fe source for alloying:** As shown in **Figure R1**, the HAADF-STEM, EELS and line intensity profiles fully prove that Fe in the support acts as an Fe source *in situ* alloying with the introduced Pt source to form ordered PtFe alloy nanoclusters loaded on the Fe_{SAs}-N-C support.

Figure R1 (Figure 1c-i). (a) HAADF-STEM image and (b) corresponding EELS analysis to verify the coexistence of PtFe and atomic level Fe in the PtFe@Fe_{SAs}-N-C electrocatalyst (the spots in the red dashed circles is ascribed to the Fe single atoms). (c) HAADF-STEM image of an individual PtFe NCs. (d) FFT pattern of figure 1c. (e) The enlarged HAADF-STEM image in the blue box in c. (f) Crystal structure of PtFe with ordered, tetragonal structure and its projection along the (001) zone axis (blue and red spheres represent Pt and Fe, respectively). (g) Intensity profile across the PtFe particle from (002) and (001) directions measured from f. The distance between the strong intensities matches with the separation between Pt atoms.

2) **Enhance metal-support interaction and inhibit the migration of alloy nanoclusters:** As shown in **Figure R2a**, the deconvolution of the Pt 4f spectra presents the main peaks for Pt⁰ along with Pt²⁺. The electron transfer from Pt to the Fe_{SAs}-N-C support causes a positive shift in the Pt 4f binding energy in PtFe@Fe_{SAs}-N-C compared to Pt/C. The FeN₄ sites embedded in the carbon substrate, behaving electronegatively, can modify the electronic structure of adjacent carbon. The resultant electron deficiency of carbon strengthens the deposition of Pt and the metal-support interaction.²² The deconvoluted N 1s spectra of Fe_{SAs}-N-C and PtFe@Fe_{SAs}-N-C (**Figure R2b**) includes a dominant graphitic N along with pyridinic N, metal N, and oxidized N. The existence of N-Fe peaks confirms the coordination between Fe and N in the carbon support. As Fe atoms are alloyed with Pt, and N-Fe bonds are broken, resulting in a significant content decrease of N-Fe and pyridinic N. The Pt L₃-edge and Fe K-edge XANES spectra also confirmed this electronic structure transition (**Figure R2c,d**).

Figure R2. (a) XPS spectra of Pt 4f for Pt/C and PtFe@Fe_{SAs}-N-C. (b) XPS spectra of N 1s for Fe_{SAs}-N-C and PtFe@Fe_{SAs}-N-C. (c) XANES spectra of PtFe@Fe_{SAs}-N-C and Pt foil: Pt L₃-edge. (d) XANES spectra of PtFe@Fe_{SAs}-N-C, Fe_{SAs}-N-C, Fe foil, and FePc: Fe K-edge.

3) **FeN₄ sites synergistically enhance the catalytic activity of PtFe@Fe_{SAs}-N-C:** As shown in **Figure R3a**, Fe_{SAs}-N-C has a certain intrinsic activity, $E_{1/2}$ can reach 0.77 V, and after the introduction of Pt to form PtFe@Fe_{SAs}-N-C catalyst, $E_{1/2}$ can reach 0.872 V. In order to further test the catalytic activity of Fe-N₄ site in PtFe@Fe_{SAs}-N-C catalyst, the anions SCN⁻ (10 mM) poisoning measurements were carried out in 0.1 M HClO₄ electrolytes (**Figure R3b**). As a result, the slight decrease of the $E_{1/2}$ reveals that the single-atom FeN₄ site around the alloy NCs also exhibits certain activity.

Figure R3. (a) ORR polarization curves of PtFe@Fe_{SAs}-N-C, Pt@N-C, Fe_{SAs}-N-C, N-C and Pt/C in O₂-

saturated 0.1M HClO₄ at a rotation rate of 1600 rpm. (b) LSV curves of PtFe@Fe_{SAs}-N-C in 0.1 M HClO₄ before and after the addition of SCN⁻.

4) **Realize charge injection effect, regulate the 5d orbital electron occupancy state of Pt sites, and synergistically optimize the ORR process:** As shown in **Figure R4a**, the magnetization and effective magnetic moment of PtFe@Fe_{SAs}-N-C gradually decrease as the applied potential reduces from 1.1 to 0.2 V vs. RHE, which is attributed to the Griffiths adsorption model of O₂ molecules and the optimization of O₂ molecules/intermediates adsorption behavior on the active Pt site. Specifically, under the driving force of electronegativity difference between Pt ($\chi = 2.28$) and Fe ($\chi = 1.83$) metal atoms, electrons are transferred from Fe to Pt, which increases the number of Pt 5d orbital occupied electrons, thus weakening the interaction between *p*-orbital for O₂ molecular and *dz*² orbital for Pt site. This is further confirmed by *operando* XANES of Pt L₃-edge and Fe K-edge. In *operando* XANES of Pt L₃-edge (**Figure R4b**), with the ORR process, that is, given the applied cathodic potential decreasing from 1.1 to 0.0 V, the white line intensity decreases successively, revealing an increase in the electron occupied state of the Pt 5d orbital. As expected, *operando* XANES of Fe K-edge showed the opposite phenomenon (**Figure R4c**), when the applied cathodic potential decreases from 1.1 to 0.0 V vs. RHE, the absorption edge continuously shifts toward high energies, which means that Fe in PtFe@Fe_{SAs}-N-C loses electrons and the valence state significantly increases during ORR process.

In conclusion, Fe_{SAs}-N-C support plays a crucial role in improving the catalytic activity and stability of PtFe@Fe_{SAs}-N-C catalyst.

Figure R4. (a) The M-T curve and effective magnetic (μ_{eff}) moment of the PtFe@Fe_{SAs}-N-C vary with the applied potential in the O₂ saturated 0.1 M HClO₄. (b) *Operando* Pt L₃-edge and (c) *Operando* Fe K-edge XANES spectra for PtFe@Fe_{SAs}-N-C in the O₂ saturated 0.1 M HClO₄.

Q2. In this paper, the authors propose that Fe injection charge into the d orbitals of Pt can optimize the spin configuration and adsorption behavior of Pt sites. However, in PtFe@Fe_{SAs}-N-C catalysts, Fe exists in both support and alloy nanoparticles. Does the Fe injection of charge into the d orbitals of Pt come from the carrier or the alloy nanoparticles?

Reply: Thank you very much for your valuable comments. The Fe that regulates the electron occupied states of the Pt 5d orbital by charge injection is derived from the support and the alloy respectively. Firstly, the work functions of the alloy (4.8 eV) and the support (4.4 eV) in the catalyst were detected by KPFM, demonstrating a trend of electrons flowing from the support to the alloy. Secondly, the quantum spin exchange interaction exists for the ferromagnetic structure of ordered PtFe alloy, and the long-range quantum spin interaction between ferromagnetic atoms delocalizes the entire catalytic structure, weakening the bonding between oxygen-containing intermediates and surface Pt sites.

We also confirm these results by DFT+U calculations. Firstly, the work function calculated for Fe-N₄-C and PtFe(111) structures is in agreement with that detected by KPFM (Figure

R5a,b). Secondly, we constructed Pt(111)/FeN₄ and PtFe(111) structures to explore the effect of Fe in the support and Fe in the alloy on the catalytic behavior of Pt sites. Results as shown in **Figure R6a-c**, the reaction energy barriers of PtFe (111) and Pt(111) /FeN₄ (both the first OH* protonation) were much smaller than those of Pt(111) and FeN₄, which alleviated the steric effect of OH* and was beneficial to ORR kinetics.

Figure R5. The work function (e_{ϕ}) of (a) PtFe (111) and (b) FeN₄.

Figure R6. (a) Gibbs free energy diagram of ORR on PtFe (111), Pt (111), and FeN₄ at $U = 0.85$ V. (b) Energy barrier for OH* protonation. (c) Gibbs free energy diagram of ORR on Pt (111)/ FeN₄, Pt (111), and FeN₄ at $U = 0.85$ V.

Q3. It is known that the biggest problem of Fe-N-C catalyst in proton exchange membrane fuel cells is Fe shedding. Whether it is taken into account in this work, please conduct necessary

analysis and proof.

Reply: Thank you very much for your thoughtful suggestion. Based on your suggestion, in order to better demonstrate that the strong metal-support interaction and the formation of ordered PtFe alloys can effectively suppress the detachment of metal Pt and Fe, we have added XPS characterization to reveal the retention rates of Pt and Fe after stability testing. As shown in **Figure R7**, after 30,000 potential cycles, for Fe_{SAS}-N-C, the amount of leached Fe is as high as 45.1%, while it is greatly reduced to 14.2% for PtFe@Fe_{SAS}-N-C. For Pt/C, the amount of leached Pt is as high as 63%, whereas it is only 12.5% for PtFe@Fe_{SAS}-N-C. The corresponding content has been added to the revised manuscript (Page 13, lines 14-16).

Figure R7. The metal retention rates of PtFe@Fe_{SAS}-N-C, Pt/C and Fe_{SAS}-N-C after 30k potential cycles (0.6-1.0 V vs. RHE) in O₂-saturated 0.1 M HClO₄.

Q4. It is good for the authors to propose the O₂-side adsorption on the Pt sites of dissociation pathway for the PtFe@Fe_{SAS}-N-C. The author should provide more sufficient evidence to directly demonstrate such mechanism, such as the corresponding calculations.

Reply: Thank you very much for your thoughtful suggestion. According to your suggestion, we calculated the O-O bond length of O₂ molecule adsorbed at the Pt site in two end-on fashion (Pauling model) and side-on adsorption configuration (Griffiths model) and the first reaction energy barrier of proton-coupled electron transfer respectively, the specific calculation results are shown in Table R1 (added to Table S6 in modified SI).

The O₂ molecule bond with Pt sites in an end-on adsorption configuration (Pauling model) over the PtFe(111) and Pt(111)/FeN₄ surface with the O-O bond elongated by 0.08 and 0.1 Å (from 1.24 to 1.32 and 1.34 Å). whereas a side-on adsorption configuration (Griffiths model) is more preferable on the PtFe(111) and Pt(111)/FeN₄ with the O-O bond elongated by 0.14 and

0.18 Å, which lowers the cleavage barrier for O-O bond and accelerates the O₂ dissociation. Notably, according to the energy barrier of the first step reduction reaction of O₂ molecules in the two adsorption configurations shown in **Table R1**, the reaction energy barrier of the Griffiths model is always smaller than that of the Pauling model, which proves that the breaking of O-O bond efficiently results in the more favorable direct 4e⁻ processes (associative pathway) than the indirect 4e⁻ processes (dissociation pathway). Corresponding analysis has been added to the revised SI (Page 37).

Table R1. The O-O bond length and first step reduction reaction energy barrier of O₂ molecules adsorbed on structures PtFe(111) and Pt(111)/FeN₄ in Pauling and Griffiths adsorption configurations

Structural model	O-O bond length		ΔG _{OOH*} (eV)	O-O bond length (Å)	
	(Å, Pauling model)			Pauling model	Griffiths model
PtFe(111)	1.32		4.326	1.38	3.448
Pt(111)/FeN ₄	1.34		3.219	1.42	2.651

Q5. Page 8, “As Fe atoms are alloyed with Pt, and N-Fe bonds are broken, resulting in a significant content decrease of N-Fe and pyridinic N.” This analysis is inconsistent with Table S4, please confirm.

Reply: Thank you very much for your thoughtful suggestion. It was our mistake to mark the wrong names of the two samples, which have been corrected in SI, and the corrected table S4 is as follows:

Table S4. The relative contents of different N derived from high-resolution XPS scans of N 1s.

Sample	NO _x (%)	N-Fe (%)	Pyridinic N (%)	Graphitic N (%)
PtFe@FeSAs-N-C	17.48	16.65	16.81	49.06
FeSAs-N-C	7.27	19.57	18.72	54.44

Q6. In Figure 6, TEM images of PtFe@FeSAs-N-C in the fuel cells after stability test results should also be provided and discussed.

Reply: Thank you very much for your thoughtful suggestion. According to your suggestion, we supplemented the TEM image after H₂-O₂ stability test, as shown in **Figure R8d-f**. After running for more than 220 h, the original morphology and elemental distribution can still be well preserved, and there is no obvious particle agglomeration, which further confirms the stability of the PtFe@FeSAs-N-C catalyst structure. The corresponding content has been added the statement that “TEM images after H₂-O₂ stability test showed (Figure S50) that the original

morphology and elemental distribution can still be well preserved, and there is no obvious particle agglomeration occurred after more than 220 h operation, which further confirmed the structural stability of PtFe@Fe_{SAs}-N-C catalyst.” in the revised manuscript (Page 21, lines 7-11.).

Figure R8. (a-c) Different magnification TEM images and elemental distribution of PtFe@Fe_{SAs}-N-C before H₂-O₂ fuel cell stability test. (d-e) Different magnification TEM images and elemental distribution of PtFe@Fe_{SAs}-N-C after H₂-O₂ fuel cell stability test.

To reviewer #3:

Xue et al. developed PtFe alloy/Fe-N-C material (PtFe@Fe_{SAs}-N-C) as an ORR catalyst for proton exchange membrane fuel cell (PEMFC) application. Extensive ex-situ and in-situ characterizations were conducted to investigate the material's structure and electronic properties. PtFe@Fe_{SAs}-N-C exhibited high activity, surpassing commercial Pt/C, and robust stability under accelerated durability testing and chronoamperometric characterization. The authors proposed an intriguing concept of the directional spin-charge injection effect to elucidate the enhanced catalytic activity. However, the justification of the authors' claims appears limited, and the data interpretation is somewhat biased.

Reply: Thank you for your recognition and valuable comments on our manuscript, which will greatly improve its quality. We have now amended the manuscript carefully based on your concerns. A list of point-to-point response is as below.

Q1. Firstly, a concern arises regarding the inconsistency in the direction of electron redistribution between PtFe nanoparticles (NPs) and the FeN₄ support. The shift of the Pt 4f XPS peak to a higher binding energy and the increase of Pt L₃-edge absorption intensity suggest “electron transfer from Pt to the Fe_{SAs}-N-C support”. Conversely, Kelvin probe force microscopy (KPFM) and DFT calculations suggest “electrons tend to flow from the Fe_{SAs}-N-C support to PtFe NCs”. The authors adopted the latter interpretation (i.e., Pt as the electron acceptor) to explain the enhanced catalytic performance. This inconsistency may stem from the dependency of Pt NPs electronic properties on NP size, as demonstrated by Yan et al. (Nat Commun 10, 4977 (2019)) and Wei et al. (Angew. Chem. Int. Ed. 2021, 60, 16622). KPFM and DFT calculations represent local characterization, focusing on specific NPs with particular sizes, whereas XPS (limit to 10 nm thickness) and XAS provide more “bulk” characterizations, probing multiple Pt NPs with varying sizes. Hence, it can be inferred that in PtFe@Fe_{SAs}-N-C, the dominant Pt species is more likely to be electron-donating than electron-accepting, contradicting the authors' claims. The authors should provide a more comprehensive discussion addressing this inconsistency.

Reply: Thank you very much for your review and thoughtful consideration. The concerns and viewpoints you have raised are consistent with the results explained in our article, that is, during the formation of PtFe@Fe_{SAs}-N-C, Pt is the electron donor and electrons are transferred to the

carrier to enhance the metal-support interaction, which is confirmed by the shift of the Pt 4f XPS peak to higher binding energy and the increase in the intensity of the Pt L₃-edge white line intensity. For KPFM testing, the purpose is to characterize the electron transfer trend of the catalyst during the ORR process, that is, the electrons on the support have a tendency to be transferred to the Pt nanoclusters, and in combination with the electronegativity difference between the two, which is much larger in Pt ($\chi = 2.28$) than in Fe ($\chi = 1.83$), the Pt becomes an electron acceptor, and electrons from Fe in the support and alloy are injected into the *d* orbitals of Pt sites, increasing the occupied state of Pt *dz*² orbitals, thereby weakening the adsorption of OH* and suppressing the site-blocking effect. This result was also verified by *operando* magnetism (real-time magnetic responses accompanying the electrochemically-driven reactions) and *operando* XAFS characterization, that is, during the ORR process, as the reaction proceeds (with a decrease in applied potential), the effective magnetic moment of PtFe@Fe_{SAs}-N-C decreases. According to the formula ($\mu_{\text{eff}} = \sqrt{n(n + 2)}$), it can be inferred that the number of unpaired electrons in the Fe 3*d* orbital decreases; The intensity of the Pt L₃-edge white line decreases, indicating an increase in the number of electrons in the Pt 5*d* orbital; At the same time, the Fe K-edge shifts towards higher energy, further confirming the loss of electrons in Fe. Therefore, during the ORR process, electrons on Fe are injected into the Pt 5*d* orbitals to optimize their electron occupancy states and interact with the orbitals of oxygen-containing intermediates. For better presentation for the reader's understanding, we have placed the characterization analysis on KPFM in the more appropriate catalytic mechanism analysis section in the revised manuscript (From line 15 on page 14 to line 2 on page 15.).

Q2.1 Secondly, the impact of 'spin charge injection' on catalytic activity remains uncertain. It is unclear whether the spin configuration of the active site is the determinant of activity or simply a dependent variable that fluctuates throughout the reaction. Additionally, it seems that orbital occupancy plays a more crucial role in determining catalytic activity compared to spin characteristics.

Reply: Thank you very much for your thoughtful consideration. ORR involves a multi-step proton coupling electron transfer process, involving spin transitions from paramagnetic triplet O₂ molecules to diamagnetic singlet oxygen-containing intermediates. The electron transfer and

orbital interactions between the catalyst and O₂ molecules/intermediates exhibit spin dependent characteristics, making reaction kinetics and thermodynamics sensitive to spin configurations. Therefore, the regulation of the spin state behavior of the catalyst will play a significant role in the ORR process. The spin configuration is determined by the orbital occupancy state, although many studies have confirmed that controlling the orbital occupancy of materials has a decisive impact on catalytic activity and stability, however, there is still a lack of studies on the changes in the spin structure of the catalytic site and the orbital-electronic interactions with the intermediate during the reaction. Therefore, the main purpose of our work is to propose and design *operando* magnetic characterization techniques combined with *operando* XAFS characterization to reveal the evolution of orbital electron interactions between Pt sites and oxygen-containing intermediates in the ORR process. Experimental results demonstrate that during the ORR process, there is a charge injection effect between Fe and Pt in the PtFe@FeSAs-N-C, which increases the electron occupancy of Pt 5d orbitals, generates additional σ^* bonds with the π orbitals of OH* intermediates, leading to a decrease in the strength of Pt-OH* bonds, suppressing site-blocking effect, and effectively promoting ORR kinetics. Of course, such research is still very limited and requires more *in-situ* characterization techniques in practical applications to reveal this evolution process and effectively promote the commercial application of low-Pt/non-Pt catalysts.

Q2.2 The authors assert that the spin charge injection effect optimizes the activation and dissociation of paramagnetic O₂ molecules (Page 18, line 381). However, both experimental and computed results suggest otherwise. O₂ temperature-programmed desorption (Figure S35) indicates weaker O₂ adsorption on PtFe@FeSAs-N-C compared to FeSAs-N-C. This weaker O₂ adsorption, resulting in less back donation of electrons from Pt 5d to O₂ π^* , leads to slower activation/dissociation, which is unfavorable for the diffusion-limited ORR. Furthermore, energy pathway analyses (Figures 5b, c, e, and S40) demonstrate that the dissociation of O₂ and the formation of O+OH* are much more thermodynamically favorable, with a significant decrease in free energy, on Pt(111) surface than on the spin-injected PtFe(111) surface.

Reply: Thank you very much for your thoughtful consideration. Based on your concerns and relevant literature reports (*Nat. Commun.* 2021, 12,1734.; *Angew. Chem.Int. Ed.* 2022, 61, e202117617.), we recognize that the relevant analysis on the adsorption of O₂ molecules by

catalytic sites needs to be corrected. Among them, for O₂-TPD, it is reasonable that the Pt site has a weaker adsorption of O₂ molecules compared to the Fe site, as the number of electrons in the Fe 3*d* orbital is less than that in the Pt 5*d* orbital. Therefore, the Fe site has a stronger adsorption of O₂ molecules, but the desorption temperature of O₂ molecules on the Pt site is very close to the Fe site. Moreover, the Pt site exhibits a stronger O₂ desorption peak intensity, further indicating its strong O₂ adsorption affinity, which is conducive to the activation of O₂. This part of the analysis has been corrected in the revised manuscript (page 16, lines 17-20): *“The Pt-O peak of PtFe@Fe_{SAs}-N-C shows a desorption temperature comparable to that of Fe-O of Fe_{SAs}-N-C, but with a stronger intensity of O₂ desorption peak, which implies that the Pt site in PtFe@Fe_{SAs}-N-C has a strong adsorption affinity for O₂, which is favorable for the activation of the O-O bond.”*

For the calculation part, we checked and found that we forgot to add U in the calculation for the Pt(111) model, while all the other structures were calculated using DFT+U. For consistency, we recalculated the 4e⁻ reaction energy barrier diagram for the Pt(111) model using DFT+U as shown below. The rate determining step (RDS) on the Pt(111) configuration is still the second OH* protonation, but the dissociation of O₂ and the formation of O+OH* are weaker than other models. The PtFe(111), Pt(111)/FeN₄, and PtFe(111)/FeN₄ constructed based on the experimental results not only have a smaller energy barrier for the RDS, but also are thermodynamically more favorable for the dissociation of O₂ and the formation of O+OH*. In summary, PtFe@Fe_{SAs}-N-C catalysts can effectively optimize the activation and dissociation of paramagnetic O₂ molecules. Corresponding content and figures have been corrected in the revised manuscript (Page 19, lines 1-10.) and SI (Figure S45), respectively.

Figure R1. (a) Gibbs free energy diagram of ORR on PtFe(111)/FeN₄, PtFe(111), Pt(111)/FeN₄, Pt(111), and FeN₄ at U = 0 V. (b) Energy barrier for OH* protonation. (c) Gibbs free energy diagram of ORR on PtFe(111)/FeN₄, Pt(111), and FeN₄ at U = 0.85 V.

Q2.3 The enhanced activity is also attributed to the facilitated protonation of OH*. Increased orbital occupancy weakens the Pt-OH* bonding, thereby promoting OH* protonation (rate-determined step). However, the effect of spin injection remains unclear. PtFe(111) and Pt(111)/FeN₄, which likely possess different spin and magnetic moments, exhibit very similar reaction energy barriers of 0.16 and 0.18 eV, respectively.

Reply: Thank you very much for your thoughtful consideration. During the ORR process, the electrons added to the Pt 5d orbitals come from the Fe 3d orbitals, which have a high spin state, and therefore, we refer to this electronic structure modulation mechanism as the "spin charge injection effect". Among them, the Fe that plays a regulatory role comes from the Fe_{SAs}-N-C support and ordered alloy, respectively. Therefore, the Pt layer adjacent to the support and the Pt in the alloy will be affected by magnetic Fe. In addition, the Pt in the alloy will also be affected by quantum spin exchange interactions between adjacent Fe magnetic layers. Based on this, we established PtFe(111) and Pt(111)/FeN₄ models in our calculations to investigate the performance of ORR. The results were as expected, with lower energy barriers for the RDE, which is more conducive to the progress of ORR.

Q2.4 Page 14, line 293, the authors attribute the decrease in magnetization and effective

magnetic moment of PtFe@Fe_{SAs}-N-C as the applied potential reduces from 1.1 to 0.2 vs. RHE, the Griffiths adsorption O₂ molecules and the optimization of O₂ molecules/intermediates adsorption behavior on the active Pt site. However, this assertion warrants further evaluation. Why does the adsorption of O₂ in an on-top configuration on a single Pt atom reduce the magnetic moment? Furthermore, it's important to note that Griffiths adsorption often leads to an associative ORR pathway rather than the dissociative pathway as claimed (Angew. Chem. 2023, 135, e202301833).

Reply: Thank you very much for your thoughtful consideration. In light of your concerns, we explain both the magnetic moment reduction and O₂ molecule adsorption mode:

As the applied potential decreases, the magnetic moment decreases, which is not directly caused by the adsorption of O₂ molecules or intermediate states, but by the charge injection effect between Fe and Pt during the reaction process. Specifically, *operando* magnetism and *operando* Pt L₃-edge XANES and Fe K-edge XANES analysis, it is the transfer of Fe 3*d* orbitals to Pt 5*d* orbitals, resulting in a decrease in the number of unpaired electrons in Fe 3*d* orbitals and a decrease in effective magnetic moment and magnetization intensity.

There are three adsorption models for O₂ molecules on sites (*J. electroanal. chem. interfacial electrochem.* 1980, 112, 231-238), as shown in **Figure R2**. In the Griffiths adsorption mode, O₂ molecules interact laterally with an active center atom, and the strong orbital interaction between O₂ molecules and active center atoms can weaken the O-O bond, leading to the dissociation and adsorption of O₂ molecules, which is beneficial for O₂ molecules to reduce to H₂O through a direct 4e⁻ pathway and avoid the production of by-product H₂O₂. In the Pauling adsorption mode, one end of the O₂ molecule points towards the active center atom. According to this adsorption method, only one oxygen atom in the O₂ molecule is effectively activated, which is not conducive to the breaking of the O-O bond. It is highly likely to undergo a 2e⁻ reaction to generate H₂O₂, corroding the catalyst and proton exchange membrane, and thus affecting the stability of the battery. In Guo et al.'s report (*Chem* 2022, 9, 1-17), it was also indicated that the Griffiths adsorption mode belongs to side end adsorption, which is more favorable for direct 4e⁻ reaction.

Figure R2. Possible adsorption modes for O₂ on catalyst.

Q3.1 The authors assert that the injection of electrons from Fe to Pt "... also restricts catalyst coalescence and retards the oxidative dissolution, diffusion, and Ostwald ripening processes" (Page 5, Lines 82-84). However, this claim lacks robust support.

Reply: Thank you very much for your thoughtful consideration. According to your prompt, we also realize that the expression of this sentence is not accurate enough, and have made corrections in the revised manuscript. In this work, due to the strong metal-support interaction and charge-directed injection effect of PtFe@FeSAs-N-C, not only electrochemical accessibility is achieved, but also the agglomeration of the catalysts is limited, and the corrosion of carbon supports, metal detachment, and Ostwald ripening process are retarded. In the revised manuscript, it should be changed to: "*It not only achieves electrochemical accessibility, but also restricts catalyst agglomeration and retards the corrosion of carbon supports, metal detachment and Ostwald ripening processes.*"

On the one hand, we can see that the particle size did not significantly increase by comparing the TEM images before and after the cycling stability test (**Figure R3a-f**); we further supplemented the TEM images of the samples after the long-term durability test of the fuel cell (as shown in **Figure R3g-i**), and found that there was no significant growth in particle size. The comparative analyses of these TEM images proved that the PtFe@FeSAs-N-C catalyst has good stability, is not easy to agglomerate during the reaction and can inhibit the Ostwald ripening process relatively well. The corresponding figures have been added to the revised SI as Figures S28 and S50.

On the other hand, carbon corrosion and Fe-N₄ demetallation have been identified as the two most likely degradation mechanisms of Fe-N-C catalysts. The nature of carbon can be investigated by the intensity ratio of the D band and G band (I_D/I_G) in Raman spectra (**Figure**

R4). The nearly equal I_D/I_G ratio indicates that before and after the cycle PtFe@Fe_{SAs}-N-C has similar graphitization degree, while for Fe_{SAs}-N-C, it is very different. In addition, after potential cycling, the ORR selectivity of PtFe@Fe_{SAs}-N-C and Fe_{SAs}-N-C shows the same change trend as the I_D/I_G ratio (**Figure R5**), that is, after 30,000 potential cycles, the H₂O₂ yield of PtFe@Fe_{SAs}-N-C is almost unchanged, while for Fe_{SAs}-N-C, it is doubled. It can be thus speculated that the enhanced stability of PtFe@Fe_{SAs}-N-C is due to the interactions between Pt and Fe atoms, and between PtFe nanoclusters and supports, which lead to reduction of Pt particle shedding and agglomeration, as well as the good retention of FeN₄ sites dispersed around the atoms of PtFe nanoclusters. To prove the hypothesis of the stability, we also performed XPS analysis on the leaching of metals during the electrochemical process. After 30,000 potential cycles, for Fe_{SAs}-N-C, the amount of leached Fe is as high as 45.1%, while it is greatly reduced to 14.2% for PtFe@Fe_{SAs}-N-C. For Pt/C, the amount of leached Pt is as high as 63%, whereas it is only 12.5% for PtFe@Fe_{SAs}-N-C (**Figure R6**). The corresponding content and figures have been added to the revised SI (Page 29-30).

Figure R3. (a-c) Different magnification TEM images and elemental distribution of PtFe@Fe_{SAs}-N-C before 30k potential cycles (0.6-1.0 V vs. RHE) in O₂-saturated 0.1 M HClO₄. (d-e) Different magnification TEM images and particle size distribution for PtFe@Fe_{SAs}-N-C after 30k potential cycles (0.6-1.0 V vs. RHE) in O₂-saturated 0.1 M HClO₄. (g-i) Different magnification TEM images and elemental distribution of PtFe@Fe_{SAs}-N-C after H₂-O₂ fuel cell stability test.

Figure R4. Raman spectra of Pt@Fe_{SAS}-N-C and Fe_{SAS}-N-C before and after 30k potential cycles (0.6-1.0 V vs. RHE) in O₂-saturated 0.1 M HClO₄.

Figure R5. H₂O₂ yields of PtFe@Fe_{SAS}-N-C and Fe_{SAS}-N-C before and after 30k potential cycles (0.6-1.0 V vs. RHE) in O₂-saturated 0.1 M HClO₄.

Figure R6. Metal (Pt and Fe) retention rate of PtFe@Fe_{SAS}-N-C, Pt/C and Fe_{SAS}-N-C after 30k potential cycles (0.6-1.0 V vs. RHE) in O₂-saturated 0.1 M HClO₄.

Q3.2 Figure S30a (HAADF-STEM of the post-ADT sample) appears to depict the aggregation of two PtFe NPs, with a size considerably larger than what is claimed (“3.35 nm without apparent agglomeration” Page 13 line 268). It is recommended to provide images with a larger view, capturing more nanoparticles to substantiate the absence of coalescence/agglomeration and Ostwald ripening.

Reply: Thank you very much for your thoughtful consideration. We supplemented the TEM images of the samples after ADT testing, as shown in **Figure R7**. The samples were able to maintain their original dodecahedral structure well after stability testing, and the average particle size of the nanoclusters remained between 2.8 and 3.2 nm, with no significant aggregation compared to the ~2.5 nm before stability testing. In the revised SI, the original Figure S28 (according to the content adjustment, the order of Figure S30 has been changed to

Figure S28) was replaced with Figure R7.

Figure S30. Different magnification TEM images and Particle size distribution for PtFe@FeSAs-N-C after 30k potential cycles (0.6-1.0 V vs. RHE) in O₂-saturated 0.1 M HClO₄.

Q3.3 It is quite surprising that the PtFe nanoparticles (NPs) can withstand the oxidative O₂-saturated acidic conditions without experiencing any dealloying or dissolution, which are typically observed in Pt-alloy ORR electrocatalysts. This is particularly intriguing considering the high oxophilicity and low redox potential of Fe. To bolster their claims, the authors should conduct further characterization studies, ICP-MS of the remaining electrolytes, ICP/EDS, XRD, and XPS of the used electrocatalyst, to elucidate any potential dissolution and oxidation phenomena.

Reply: Thank you very much for your thoughtful consideration. Based on your suggestion, we performed additional tests of ICP-MS of the remaining electrolyte and the samples after the stability test, Raman, and XRD. As shown in **Table R1**, the decay rates of the metal elements in the samples before and after the cycling were measured by ICP-MS and XPS techniques, and both of them were significantly lower for the PtFe@FeSAs-N-C sample than for the FeSAs-N-C.

The phase of the catalyst was measured by XRD (**Figure R8a**). After stability testing, the PtFe@Fe_{SAs}-N-C sample was still composed of carbon and PtFe phases, with no new phases appearing and no significant changes in diffraction peak intensity. Only the (200) and (002) crystal planes of carbon were observed in the Fe_{SAs}-N-C sample. Furthermore, by comparing the Raman spectra of the samples before and after stability (**Figure R8b**), it was shown that the PtFe@Fe_{SAs}-N-C sample can still maintain the carbon defect structure well before and after cycling without being destroyed. Corresponding analysis has been added to pages 13, lines 10-19 of the revised manuscript, and corresponding figures and tables have been added to the revised SI as Figures S31 and Table S6.

Table R1. The Pt and Fe contents indifferent samples as determined by XPS and ICP.

Sample	element	ICP-MS of the remaining electrolytes, (wt %)	ICP-MS (decay rate, %)	XPS (decay rate, %)
PtFe@Fe _{SAs} -N-C	Pt	1.212	7.5	12.5
	Fe	1.211	9.46	14.2
Fe-N-C	Fe	1.891	38.1	45.1

Figure R8. (a) PXRD and (b) Raman spectra of PtFe@Fe_{SAs}-N-C and Fe_{SAs}-N-C before and after accelerated durability testing.

Q3.4 Optionally, the authors may consider calculating the coherence/bonding energy of PtFe to elucidate its structural stability. Additionally, conducting electron localization function (ELF) analyses could help illustrate the “strong metal-support interaction” between PtFe nanoparticles (NPs) and the FeN₄ support, offering a mechanistic understanding of the material's robust stability.

Reply: Thank you very much for your thoughtful suggestion. Based on your suggestion, we

first calculated the adsorption energy of PtFe on FeN₄ support, and the results showed that the adsorption energy was -0.792 eV, proving the stability of the PtFe/FeN₄ structure. Secondly, we calculated the electronic localization function (ELF) and charge density difference of the PtFe/FeN₄ structure, as shown in **Figure R9**. It is evident that there is charge aggregation between the PtFe alloy and the FeN₄ support, indicating a strong metal carrier interaction between the PtFe nanoparticles (NPs) and the FeN₄ support. Corresponding analysis has been added to the revised manuscript (Page 19, lines 1-6.).

Figure R9. (a) Electronic localization function (ELF) and (b) Charge density difference of the PtFe/FeN₄ structure.

Q3.5 It is well-known that the start-up/shutdown procedure in fuel cells can induce potential excursions, leading to catalytic deterioration that is more severe than under normal load condition (0.6-1.0 vs. RHE). Optionally, the authors may further assess the stability of materials under start-up/shutdown conditions (1-1.5 vs. RHE) following the DOE-suggested protocol.

Reply: Thank you very much for your thoughtful suggestion. Based on your suggestion, we refer to DOE-suggested protocol (Nat. Catal. 2022, 5, 455-462.) and use the fast triangular wave voltage cycling method to evaluate the durability of PtFe@FeSAs-N-C catalyst between 1.0 V-1.5 V, and compare it with commercial Pt/C, as shown in **Figure R10**.

As shown in **Figure R10a**, the initial peak power density of PtFe@FeSAs-N-C and Pt/C were 1.24 W cm⁻² and 1.38 W cm⁻², respectively. After 5,000 high potential cycles, the peak power density of PtFe@FeSAs-N-C and Pt/C decreased by 14.5% and 31.5%, respectively. The activity decay rate of PtFe@FeSAs-N-C was significantly lower than the DOE target ($\leq 40\%$, *DOE Technical Targets for Polymer Electrolyte Membrane Fuel Cell Components* <https://energy.gov/eere/fuelcells/doe-technical-targetspolymer-electrolyte-membrane-fuel-cell-components> (US DOE, 2016)). The mass activity (MA) of PtFe@FeSAs-N-C and Pt/C before

durability testing were 0.75 and 0.16 A mg_{Pt}⁻¹, respectively. After 5,000 high potential cycles, they decreased to 0.64 and 0.10 A mg_{Pt}⁻¹, respectively. The MA decay rate of PtFe@Fe_{SAs}-N-C was only 14.7%, far below the DOE target ($\leq 40\%$, *DOE Technical Targets for Polymer Electrolyte Membrane Fuel Cell Components* <https://energy.gov/eere/fuelcells/doe-technical-targets-polymer-electrolyte-membrane-fuel-cell-components> (US DOE, 2016); and *Nat. catal.* 2020, 6, 503-512.). Combined with the results of the stability evaluation in the range of 0.6-0.95 V, which has been conducted previously, the excellent stability of PtFe@Fe_{SAs}-N-C is fully demonstrated. The corresponding content and figures have been added to the revised SI (Page 47-48).

Figure R10. H₂/O₂ fuel cell polarization (left axis) and power density (right axis) plots of (a) Pt/C and (b) PtFe@Fe_{SAs}-N-C cathode before (BOT) and after 5,000 potential cycles between 1.0 and 1.5 V. (c) Peak power density and (d) Mass activity (MA).

Q4. The density functional theory (DFT) analysis appears to lack comprehensiveness. It is unclear why the authors did not explore the model of PtFe(111)/FeN₄, which would reflect the structure of PtFe@Fe_{SAs}-N-C.

Reply: Thank you very much for your thoughtful suggestion. Based on your suggestion, we

have constructed the PtFe(111)/FeN₄ model and supplemented the calculation of the 4e⁻ reaction energy barrier diagram for this model, as shown in **Figure R11**. For the PtFe(111)/FeN₄ structure, the RDS is also the protonation of the first OH*, and as confirmed experimentally, the RDS energy barrier is the lowest of all structures at only 0.033 eV, proving that this structure is more favorable for 4e⁻ ORR. The corresponding content has been added to the revised manuscript (Page 19, lines 1-10.).

Figure R11. (a) Gibbs free energy diagram of ORR on PtFe(111)/FeN₄, PtFe(111), Pt(111)/FeN₄, Pt(111), and FeN₄ at $U = 0$ V. (b) Energy barrier for OH* protonation. (c) Gibbs free energy diagram of ORR on PtFe(111)/FeN₄, Pt(111), and FeN₄ at $U = 0.85$ V.

Q5. Page 11, line 211, the statement “the decrease of the magnetic moment of Fe atoms with low Pt loading can be attributed to the spin charge injection effect between Fe and Pt sites...” lacks clarity regarding how spin redistribution could reduce the average magnetic moment. Alternatively, the decrease in magnetic moment may stem from the decrease in Fe content in PtFe@FeSAs-N-C compared to FeSAs-N-C (Table S2).

Reply: Thank you very much for your thoughtful suggestion. The decrease in the magnetic moment of PtFe@FeSAs-N-C may be due to the following two reasons: firstly, as you mentioned, the slight decrease in Fe content; secondly, in the formation process of PtFe alloy, part of the single atom Fe in the support acts as a source of Fe to alloy with Pt, and the magnetic moment decreases with the formation of the alloy, whereas the ordered PtFe alloy formed has a certain

degree of magnetism due to the quantum spin-exchange interactions which provide spin-electron-transfer channels for the spin-dependent electrocatalytic reaction. Therefore, the atomic magnetic moment of PtFe@Fe_{SAs}-N-C has a slight decrease with respect to that of Fe_{SAs}-N-C.

Q6. Importantly, the authors should not assume that the electron donation from Fe sites in PtFe NCs and Fe_{SAs}-N-C impose a $t_{2g}^6e_g^3$ electron configuration on Pt atoms. The orbital occupation of t_{2g} and e_g orbitals in Pt should be determined through fitting analysis of Pt L₃-edge XANES.

Reply: Thank you very much for your thoughtful suggestion. According to your suggestion, we performed valence state fitting for both *ex-situ* and *operando* Pt L₃-edge XANES, as shown in **Figure R12c**, the valence states at the initial high potential (1.1 V) and *ex-situ* are basically the same, and the valence state of Pt decreases as the ORR proceeds (the externally applied potential decreases). The transient valence states of Pt are close to +1 valence at both 0.7 and 0.5 V of applied potential (1.02 and 0.91, respectively), at which time the electron occupancy state of the Pt 5d orbital can only be in the $t_{2g}^6e_g^3$ configuration. **Figure R12** was added to the revised SI as Figure S37.

Figure R12. (a) XANES spectra of PtFe@Fe_{SAs}-N-C, PtO₂ and Pt foil: Pt L₃ edge. (b) *Operando* Pt L₃-edge XANES spectra for PtFe@Fe_{SAs}-N-C in 0.1 M HClO₄. (c) Liner fitting for Pt valences in PtFe@Fe_{SAs}-N-C derived from corresponding Pt L₃-edge XANES spectra.

Q7. Figure 1b inset, the particle size distribution of PtFe NPs was analyzed in a very narrow range (2-3 nm), leading to the claim of “excellent size uniformity of ~2.46 nm”. The TEM and HAADF-STEM images (Figure 1c, S5, and S6), however, reveal the existence of numerous nanoparticles with much larger sizes (5-10 nm) and lower uniformity.

Reply: Thank you very much for your thoughtful consideration. The excessively large particle size may be due to localized selection. In order to more accurately identify the particle size of the alloy nanoclusters in the PtFe@Fe_{SAs}-N-C catalysts, the average particle size before and

after the stability test in this work was statistically analyzed by TEM. As shown in **Figure R13**, the average particle size of the alloy nanoparticles before cycling was ~ 2.46 nm.

Figure R13. Different magnification TEM images and Particle size distribution of PtFe@FeSAs-N-C.

Q8. Using simulated STEM images (Figure 1e, Figure S31b) as evidence to study lattice structure should be approached with caution. The STEM reconstruction algorithm may introduce significant artifacts, especially when dealing with blurry HAADF-STEM images (Figure 1c, S31a). To ensure accuracy, the authors may consider providing clearer HAADF-STEM or ACTEM images.

Reply: Thank you very much for your thoughtful consideration. The STEM image in the manuscript is obtained by applying inverse Fourier transform to the original image, and does not alter the lattice structure of the catalyst itself or introduce artifacts. But for better presentation, we replaced Figure 1e,f with the original HAADF-STEM with the same crystal structure, as shown in **Figure R14**.

Figure R14 (Figure 1e,f). (e) HAADF-STEM image of an individual PtFe NCs. (f) FFT pattern of figure 1e.

Q9. Figure S15 and Table S5 (^{57}Fe Mössbauer spectroscopy), the percentage of iron metal (α and γ -Fe) appears to be relatively high, exceeding 30%. The authors should explain the formation of these components and elucidate whether they have any impact on the catalytic performance.

Reply: Thank you very much for your thoughtful consideration. ^{57}Fe Mössbauer studies of Fe alloys, such as Pt-Fe (*J. Catal.* 1973, 29, 278-291 and *Nat. Catal.* 2022, 5, 503-512.), Ru-Fe (*J. Mol. Catal.* 1984, 25, 285-293 and Pd-Fe (*J. Catal.* 1976, 43, 18-33.) suggested that Fe was zero valent in the alloys. Bartholomew and Boudart (*J. Catal.* 1973, 29, 278-291.) observed that the Mössbauer absorption probability of surface Fe atoms in the alloy was substantially the same as those in the bulk, especially for nanosize Pt-Fe particles. Thus, the iron metal component in PtFe@FeSAs-N-C is largely due to Pt-Fe alloys, whereas the doublets are from nitrogen-coordinated Fe single atoms.

Q10. In PEMFC characterization, it is unclear why the authors did not use the same Pt loading for Pt/C and PtFe@FeSAs-N-C.

Reply: Thank you very much for your thoughtful consideration. The dispersion of the catalytic layer is closely related to the catalytic performance, and we have derived their best performance by modulating the dispersion of the catalytic layer for commercial Pt/C (20%) loading of 0.2 mg cm⁻² and PtFe@FeSAs-N-C loading of 0.12 mg cm⁻². Moreover, our measured commercial Pt/C performance is comparable to that reported in the current literature, thus, the methodology for evaluating catalyst fuel cells in this work is reliable.

Q11. Minor comments

Q11-1. Page 3, lines 56-57, the mention of "...introducing new variables (second adsorption site)." may not be entirely relevant, as the O₂ adsorption is the on-top adsorption (Griffiths model).

Reply: Thank you very much for your thoughtful consideration. This section summarizes the current solution strategies, in which the introduction of new variables such as the second adsorption site to form a double site to realize the ORR of O₂ molecules in a bridge adsorption mode is a commonly used and effective strategy to promote the activation of the O-O bond and improve the 4e⁻ selectivity. We reviewed the earliest articles proposing O₂ adsorption modes, there are three adsorption models for O₂ molecules on sites (*J. electroanal. chem. interfacial electrochem.* 1980, 112, 231-238.), as shown in **Figure R2**. In the Griffiths adsorption mode, O₂ molecules interact laterally with an active center atom, and the strong orbital interaction between O₂ molecules and active center atoms can weaken the O-O bond, promote the dissociation of O₂ molecules, which is beneficial for O₂ molecules to reduce to H₂O through a

direct $4e^-$ pathway and avoid the production of by-product H_2O_2 . In Guo et al.'s report (*Chem* 2022, 9, 1-17.), it was also indicated that the Griffiths adsorption mode belongs to side end adsorption, which is more favorable for direct $4e^-$ reaction.

Figure R2. Possible adsorption modes for O_2 on catalyst.

Q11-2. Figure S2 (PXRD) unveiled a shift in the diffraction peaks of $PtFe@Fe_{SAs}-N-C$ towards lower 2θ (e.g., (001) and (111)), suggesting the presence of tensile strain. The authors should comment on the impact of this strain on the material's physicochemical properties and catalytic performance.

Reply: Thank you very much for your thoughtful suggestion. Strain is a universal phenomenon, which is a localized relative deformation of an object under the action of external forces and non-uniform temperature field and other factors. As early as 1998, professor Nørskov has revealed through theoretical calculations that the strain of metallic materials is closely related to the d -band center, and the difference of the d -band center (E_d) with respect to the Fermi energy level can well reflect the adsorption energy of adsorbates on the catalyst surface (*Phys. Rev. Lett.* 1998, 81, 2819-2822.). The present work focuses on the modulation of the active center Pt $5d$ orbital electron occupancy state based on the spin charge injection effect using low electronegativity Fe during the ORR process, revealing the orbital-electron interactions between the intermediate state and the Pt sites, which is more specific and in-depth than the d -band center theory.

Q11-3. Page 4, line 61, “proton coupling and electron transfer steps” should be “proton-coupled electron transfer steps”.

Reply: Thank you very much for your review. Based on your suggestions, we have made the correct modifications in the revised manuscript.

Q11-4. Figure 1b caption, “the coexistence with FeN_4 sites” should be removed as the atomic

FeN₄ cannot be observed in TEM image.

Reply: Thank you very much for your careful consideration. In the revised manuscript, we have removed the word "the coexistence with FeN₄ sites" from the Figure 1b caption.

Q11-5. The caption of Figure 1d inset should be provided.

Reply: Thank you very much for your careful consideration. In the revised manuscript, we have added the explanation of the caption in Figure 1d.

Q11-6. Figure 1f (FFT pattern), the authors may consider providing a clearer image.

Reply: Thank you very much for your careful consideration. Based on your suggestion, we have replaced Figures 1e,f to present them accurately and clearly to the readers.

Figure R14 (Figure 1e,f). (e) HAADF-STEM image of an individual PtFe NCs. (f) FFT pattern of figure 1e.

Q11-7. Figure 1h, "d(002) should be "d(111)" as planes (002) and (001) are parallel.

Reply: Thank you very much for your careful consideration and correction. Based on the lattice and exposed crystal planes in Figures 1e,f, we corrected the marks in Figure 1h, as shown in the figure below.

Figure R15 (Figure 1). (e) HAADF-STEM image of an individual PtFe NCs. (f) FFT pattern of figure 1e. (g) The enlarged STEM image in the blue box in e. (h) Crystal structure of PtFe with ordered, tetragonal structure and its projection along the (001) zone axis (blue and red spheres represent Pt and Fe, respectively).

Q11-8. Table S4, the row label seems to be incorrect, as the contents of N-Fe and pyridinic N

in PtFe@Fe_{SAs}-N-C are higher than those in Fe_{SAs}-N-C.

Reply: Thank you very much for your comprehensive consideration. It was our mistake to mark the wrong names of the two samples, which have been corrected in SI, and the corrected table S4 is as follows:

Table S4. The relative contents of different N derived from high-resolution XPS scans of N 1s.

Sample	NO _x (%)	N-Fe (%)	Pyridinic N (%)	Graphitic N (%)
PtFe@Fe _{SAs} -N-C	17.48	16.65	16.81	49.06
Fe _{SAs} -N-C	7.27	19.57	18.72	54.44

Q11-9. Figure 3d, “inset: the absorbance value of the solution at 417 nm” is missing.

Reply: Thank you very much for your comprehensive consideration. We added an illustration for the corresponding caption, as shown in **Figure R16**, and replaced the original Figure 3d.

Figure R16. UV-Vis absorption spectra of 0.1 M HClO₄ solution of ABTS+H₂O₂ with PtFe@Fe_{SAs}-N-C, Fe_{SAs}-N-C catalyst and no catalyst. Inset: the absorbance value of the solution at 417 nm.

Q11-10. Page 10 line 205, is it worth mentioning the presence of “one eg electron ($t_{2g}^4 e_g^1$), the medium-spin Fe(III)N₄” considering that Fe_{SAs}-N-C contains significantly more medium-spin Fe(III)N₄ than PtFe@Fe_{SAs}-N-C.

Reply: Thank you very much for your comprehensive consideration. Based on your prompt, after careful consideration, we believe that this expression is not necessary, so we will delete it.

Q11-11. Figure 6b caption, “4 mg_{Pt} cm⁻² catalyst loading for Fe_{SAs}-N-C” should be “4 mg cm⁻² catalyst loading for Fe_{SAs}-N-C”.

Reply: Thank you very much for your comprehensive consideration. We have corrected "4 mg_{Pt} cm⁻²" to "4 mg cm⁻²".

Q11-12. Page 19 line 390, the meaning of “combining with the low reaction energy barrier of Pt(111)/FeN₄ can effectively improve the ORR kinetics of PtFe@Fe_{SAs}-N-C and make it have

superior stability” is unclear. Why the low energy barrier in the Pt(111)/FeN₄ would benefit the kinetics of PtFe@FeSAs-N-C?

Reply: Thank you very much for your review. The reaction energy barrier here refers to the protonation of OH* in the rate determining step (ΔG_{OH^*}). The lower ΔG_{OH^*} , the weaker the site-blocking effect, which can promote protonation and desorption and accelerate the reaction kinetics. We have revised the expression of this sentence in the revised manuscript to “Combining with the low RDS OH* protonation energy barrier of Pt (111)/FeN₄ can effectively improve the ORR kinetics of PtFe@FeSAs-N-C and make it have superior stability.”

Q11-13. Section Methods/Electrocatalytic measurement, the mentions of “alkaline medium”, “0.1 M KOH” and the potential range “between 0.15 V and 1.05 V vs. RHE for 10000 cycles” in stability test may be typos.

Reply: Thank you very much for your review and reminder. It is indeed our mistake. We have corrected it in the revised manuscript to correspond to the testing conditions in the 0.1 M HClO₄ electrolyte used in this work.

Q11-14. The authors should also revise the manuscript to correct typos and grammatical errors. Importantly, any claims that are not been derived from authors’ results should be cited. For instance, the source of 2025 DOE target should be cited.

Reply: Thank you very much for your review. We have thoroughly reviewed and revised the entire text to avoid typing and grammar errors as much as possible. At the same time, source citations were made regarding the location of the first mention of the 2025 DOE target in the article.

REVIEWERS' COMMENTS

Reviewer #1 (Remarks to the Author):

The authors have revised the manuscript based on the comments. They also made additional experiments for improving the manuscript. All the concerns I raised have been well replied. Thus, I recommend it for the publication in Nature Commun. in present form.

Reviewer #2 (Remarks to the Author):

The authors addressed my concerns and questions. It is publishable now.

Reviewer #3 (Remarks to the Author):

It can be published.